# In Model, In Vitro and In Vivo Killing Efficacy of Antitumor Peptide RDP22 on MUG-Mel2, a Patient Derived Cell Line of an Aggressive Melanoma Metastasis

**DOI:** 10.3390/biomedicines10112961

**Published:** 2022-11-17

**Authors:** Maximiliane Wußmann, Florian Kai Groeber-Becker, Sabrina Riedl, Dina Alihodzic, Daniel Padaric, Lisa Gerlitz, Alexander Stallinger, Bernadette Liegl-Atzwanger, Dagmar Zweytick, Beate Rinner

**Affiliations:** 1Translational Center Regenerative Therapies TLC-RT, Fraunhofer Institute for Silicate Research ISC, Neunerplatz 2, 97080 Würzburg, Germany; 2Department Tissue Engineering & Regenerative Medicine (TERM), University Hospital Würzburg, Röntgenring 11, 97080 Würzburg, Germany; 3Institute of Molecular Biosciences, University of Graz, Humboldtstrasse 50/III, A-8010 Graz, Austria; 4BioHealth, University of Graz, Universitätsplatz 3, A-8010 Graz, Austria; 5BioTechMed-Graz, Mozartgasse 12/II, A-8010 Graz, Austria; 6Division of Biomedical Research, Medical University of Graz, Stiftingtalstrasse 24, A-8010 Graz, Austria; 7Diagnostic and Research Institute of Pathology, Medical University of Graz, Neue Stiftingtalstrasse 6, A-8010 Graz, Austria

**Keywords:** melanoma metastases, NRAS mutation, antitumor peptide, tumor model systems, phosphatidylserine

## Abstract

The host defense derived peptide was assessed in different model systems with increasing complexity employing the highly aggressive NRAS mutated melanoma metastases cell line MUG-Mel2. Amongst others, fluorescence microscopy and spectroscopy, as well as cell death studies were applied for liposomal, 2D and 3D in vitro models including tumor spheroids without or within skin models and in vivo mouse xenografts. Summarized, MUG-Mel2 cells were shown to significantly expose the negatively charged lipid phosphatidylserine on their plasma membranes, showing they are successfully targeted by RDP22. The peptide was able to induce cell death in MUG-Mel2 2D and 3D cultures, where it was able to kill tumor cells even inside the core of tumor spheroids or inside a melanoma organotypic model. In vitro studies indicated cell death by apoptosis upon peptide treatment with an LC_50_ of 8.5 µM and seven-fold specificity for the melanoma cell line MUG-Mel2 over normal dermal fibroblasts. In vivo studies in mice xenografts revealed effective tumor regression upon intratumoral peptide injection, indicated by the strong clearance of pigmented tumor cells and tremendous reduction in tumor size and proliferation, which was determined histologically. The peptide RDP22 has clearly shown high potential against the melanoma cell line MUG-Mel2 in vitro and in vivo.

## 1. Introduction

Cutaneous melanoma is a highly aggressive malignancy with growing incidence and limited therapy strategies [1,2]. Cutaneous melanomas are classified into four subtypes: (i) mutation of the B-Raf proto-oncogene serine/threonine kinase (BRAFmut), (ii) mutation of the NRAS proto-oncogene GTPase (NRASmut), (iii) mutation of neurofibromin 1 (NF1mut), and (iv) none of the three, triple BRAF/NRAS/NF1 wild-type (WT) [3,4,5]. The development of targeted therapies and immunotherapies has led to significant improvement in overall survival, particularly in non-NRAS-mutated melanoma (BRAF). Unfortunately, effective targeted therapies are lacking in patients with NRASmut melanomas due to emerging resistance and high aggressiveness, generally resulting in a poorer prognosis [6]. As a result, to date, there is an unmet clinical need for sustained effective therapeutic strategies for NRASmut melanoma patients. Considerable efforts have been made to inhibit the MAPK pathway alone or in combination with other drugs. New classes of inhibitors such as the proteolysis targeting chimera (PROTAC), which involves the use of novel mono-, dual-, or even triple-targeting therapies, or mRNA vaccines that can stimulate a targeted immune response against personalized tumor antigens are being adequately investigated [6,7]. Nevertheless, intrinsic and acquired resistance is likely to occur, and sometimes toxicities and side effects are massively burdensome for the patient. For the majority of advanced melanoma patients, new classes of pharmacological agents need to be considered to achieve a long and well-tolerated PFS (progression-free survival). Combined with the highly pigmented melanogenic cell line MUG-Mel2 derived from a cutaneous metastasis of an aggressive NRAS p Q61R-mutated melanoma [8] and innovative therapeutic options from the field of peptides, we demonstrate a new potential therapy [9,10,11,12,13,14].

Cationic antitumor peptides derived from the human host defense peptide Lactoferricin [15] have been shown to selectively kill cancer cells of different types including malignant melanoma and their metastases [9,11,12,13,14] in 2D and 3D in vitro and in vivo [16] by targeting the negatively charged lipid phosphatidylserine (PS), specifically exposed by the outer leaflet of the plasma membrane of cancer cells [17,18]. PS exposure was not only shown for different cancer types such as melanoma, glioblastoma, prostate and renal cancer compared to absence in non-neoplastic control cells, but it was further demonstrated that PS exposure was not attributed to ongoing apoptosis [17]. One of these peptides, RDP22 (R-DIM-P-LF11-322, PCT/EP2014/050330; US 14/760,445; EP 14700349.5), exhibits selective activity against the cancer cell marker PS in model systems and cytotoxicity against melanoma without significant effect on non-neoplastic cells as melanocytes or fibroblasts at antitumor-active concentrations [9,13,14].

For enhanced output of in vitro experiments, before in vivo studies, a number of 3D studies were applied such as tumor spheroids and second organotypic melanoma models. Accrued advantages of the latter such as the recently published mOS-REp [19] are traced back to the fact that these 3D models bridge the gap between the simplicity of 2D cell culture or spheroids lacking a microenvironment and the complexity of full-thickness skin equivalents and animal models [20,21]. The models in which the tumors mature within a physiologically differentiated epidermis enable the investigation of melanomagenesis, melanoma treatment and cellular or molecular crosstalk between the tumor and its surrounding skin cells.

The study investigates the sensitivity of the aggressive NRAS mutated melanoma MUG-Mel2 for treatment with antitumor peptide RDP22. Application of elevated model systems up to 3D tumor spheroids and organotypic melanoma models can further help to significantly reduce the number of animal studies within the preclinical phases of the development of anti-melanoma therapies.

## 2. Materials and Methods

### 2.1. Materials

The peptide R-DIM-P-LF11-322 (RDP22) (PFWRIRIRRPRRIRIRWFP-NH2, M = 2677.26 g/mol) and the fluorescently labeled peptide ((5–6)-FAM-) RDP22 ((5–6)-FAM-) PFWRIRIRR-P-RRIRIRWFP-NH_2_, M = 3035.7 g/mol) were purchased from PolyPeptide (San Diego, CA, USA). The purity was >96% as determined by RP-HPLC. The peptides were dissolved in acetic acid (0.1%, *v*/*v*) at concentrations of 3 mg/mL for model and in vitro experiments and in Dulbecco’s Phosphate Buffered Saline, DPBS (no calcium, no magnesium; Gibco^®^, Thermo Fisher Scientific, Waltham, MA, USA) buffer at concentrations of 2 mg/mL before in vivo experiments. The peptide concentration was determined by measurement of UV-absorbance of tryptophan at a wavelength of 280 nm using a NanoDrop photometer (ND 1000, Peqlab, VWR International, Inc. Erlangen, Germany). All peptide stocks were stored at 4 °C till use. The lipid 1,2-dipalmitoyl-sn-glycero-3-phospho-L-serine (sodium salt) (DPPS) was purchased from Avanti Polar Lipids, Inc. (Alabaster, AL, USA).

### 2.2. Model Studies

#### 2.2.1. Liposomal Preparations

Lipid films of 5 mg DPPS (microscopy) were prepared by drying respective amounts of lipid under a stream of nitrogen and storage in a vacuum overnight. The dry lipid film was then dispersed in respective volumes of phosphate-buffered saline (PBS, 20 mM NaPi, 130 mM NaCl, pH 7.4) and hydrated at temperatures well above the gel-to-fluid phase transition of the respective phospholipid under intermittent vigorous vortex-mixing, as described previously [14]. The lipid concentration was 0.25–0.50 mM for microscopic inspection. DPPS hydration was carried out at 65 °C for two hours by vortexing every 15 min. The fully hydrated samples were stored at room temperature until measurement.

#### 2.2.2. Microscopy Studies on Liposomes

For microscopic studies, 250–500 µg liposomes of DPPS were centrifuged at 10,000× *g* for 10 min at room temperature. The liposomal pellet was resuspended in Annexin binding buffer (ABB) and stained with Annexin V Alexa Fluor 350 conjugate according to the manufacturer’s protocol (Molecular Probes Inc., Eugene, OR, USA). PS exposure by Annexin V binding was visualized with a Leica DMI6000 B with IMC using a Leica DFC360 FX camera and AF 6000 software (Leica Microsystems, Vienna, Austria) (excitation wavelength 346 nm; emission wavelength 442 nm). For investigation of potential PS-peptide co-localization prior to PS-staining, incubation of 0.25–0.50 mM DPPS liposomes with 2 μM fluorescently labeled RDP22, ((5-6)-FAM-) RDP22 (excitation wavelength λ_ex_ = 495 nm; emission wavelength λ_em_ = 519 nm) for 10 min was performed.

### 2.3. Cell Culture

Melanoma cells MUG-Mel2 (p33) were cultured in RPMI 1640 medium with GlutaMAX™ (Gibco^®^, Thermo Fisher Scientific, Waltham, MA, USA) supplemented with 10% FBS (fetal bovine serum; Gibco^®^, Thermo Fisher Scientific, Waltham, MA, USA). Normal human dermal fibroblasts NHDF (PromoCell Inc., Heidelberg, Germany) were used as non-neoplastic controls. NHDF were cultured in Fibroblast Growth Medium 2 (PromoCell Inc., Heidelberg, Germany). At 90% confluence cells were passaged with accutase (Gibco^®^, Thermo Fisher Scientific, Waltham, MA, USA). All cells and spheroids were kept in 5% CO_2_ at 37 °C and periodically checked for mycoplasma. Cell line authentication was performed by using the Power Plex^®^ 16 HS System (Promega, Madison, WI, USA).

### 2.4. PS Exposure

PS exposure of MUG-Mel2 was determined using the Annexin V Alexa 488 apoptosis detection kit (Molecular Probes Inc., Eugene, OR, USA). Briefly, 10^5^ cells were seeded into black 96-well plates and incubated O/N at 5% CO_2_ and 37 °C. Subsequently, cells were washed once with Annexin V binding buffer and stained with 5 µL Annexin V-Alexa 488 and 2 µL PI (50 µg/µL) for 5 min at room temperature in the dark. Cells were washed again and fluorescence (Annexin V Alexa Fluor 488, λ_ex_ = 488 nm and λ_em_ = 530 nm; PI, λ_ex_ = 536 nm and λ_em_ = 617 nm) was recorded immediately using the GloMax^®^ Discover Microplate Reader (Promega, Madison, WI, USA).

### 2.5. Two-Dimensional In Vitro Experiments

#### 2.5.1. Analysis of Cell Death—Fluorescence Spectroscopy—PI Uptake

Cells were collected, resuspended in respective media (see cell culture) and diluted to a concentration of 10^6^ cells/mL. Aliquots of 10^5^ cells in suspension were incubated in presence of respective concentrations of peptide (0 µM up to 20 µM) for up to 8 h in the presence of propidium iodide PI (2 µL/10^5^ cells of 50 µg/mL, Molecular Probes Inc., Eugene, OR, USA) at room temperature in black 96-well plates. PI-uptake was measured after 0, 1, 2, 4 and 8 h using the GloMax^®^ Discover Microplate Reader (Promega, Madison, WI, USA). Cytotoxicity was calculated from the percentage of PI-positive cells in media alone (P_0_) and in the presence of peptide (P_X_) (Equation (1)). Triton-X-100 (10%) was used to determine 100% of PI-positive cells (P_100_).
(1)$$\%\;\mathrm{PI}-\mathrm{uptake}=\frac{100*\left(P_{x}-P_{0}\right)}{\left(P_{100}-P_{0}\right)}$$

Excitation and emission wavelengths were 536 nm and 617 nm, respectively. Each experiment was repeated at least three times.

#### 2.5.2. Visualization of Cell Death 2D—Fluorescence Microscopy—PI Uptake

Cells (1–5 × 10^4^) were seeded on Ibidi µ-Slide 8 wells (ibidi GmbH, Martinsried, Germany) and grown in 300 µL media for 2–3 days to a confluent layer. The peptide was added at different concentrations and incubated for 8 h at 37 °C and 5% CO2. Propidium iodide PI (2 µL/well of 50 µg/mL, Molecular Probes Inc., Eugene, OR, USA) was added to the wells and incubated for another 5 min in the dark. Experiments were performed with a Leica DMI6000 B with IMC using a Leica DFC360 FX camera and AF 6000 software (Leica Microsystems, Vienna, Austria). Excitation and emission wavelengths were 536 nm and 617 nm, respectively. PI can only enter cells and intercalate with the DNA upon potential peptide-induced membrane damage. For the visualization of the interaction of the peptide with PS and induced cell death, MUG-Mel2 cells seeded and grown for 2–3 days on Ibidi µ-Slides were incubated with 10 μM ((5-6)-FAM-) RDP22 for up to 6 h. Cells were then stained and examined with Annexin V Alexa Fluor 350 conjugate and PI as described in 2.2.2. For the visualization of the interaction of RDP22 with non-malignant control cells, NHDF were analyzed on Ibidi µ-Slides upon incubation with 10 μM ((5-6)-FAM-) RDP22 for up to 4 h and stained with Annexin V-Alexa 488 and PI as described in 2.4.

#### 2.5.3. Apoptosis/Necrosis Assay

To assess whether the peptide induces apoptosis or necrosis, the RealTime-Glo™ Annexin V Apoptosis (luminescence) and Necrosis (fluorescence) Assay from Promega (Promega, Madison, WI, USA) were used. Briefly, cells were seeded at 10^5^ cells/100 µL in a white 96-multiwell plate with clear bottom and incubated overnight at 5% CO2 and 37 °C. The reagent stock solution (1000×) was diluted in medium according to the manufacturer’s protocol. Before the measurement, 100 µL of the reagent solution was added to the cell suspension. For the fluorophore, excitation and emission wavelengths were 490 nm and 525 nm, respectively. Before peptide addition, the signals at time zero were measured with the GloMax^®^ Discover Microplate Reader (Promega, Madison, WI, USA). Thereupon, the cells were treated with peptides at different concentrations (5–20 µM). Apoptosis and necrosis signals were measured every 30 min for 6 h.

#### 2.5.4. Caspase-3/7 Assay

To further assess whether cells undergo apoptosis the Caspase-Glo^®^ 3/7 Assay (Promega, Madison, WI, USA) was used; 2 × 10^5^ cells/mL were seeded in a white 96-multiwell plate with clear bottom and grown for 48 h at 5% CO_2_ and 37 °C. Peptide at 0 and 10 µM was added for incubation over 4 h. Caspase solution was added with a 1:1 ratio upon 30 min at RT. Luminescence was measured with a GloMax^®^ Discover Microplate Reader (Promega, Madison, WI, USA). Mean values of caspase 3/7 activity were analyzed as a multiple of non-peptide treated cells. Three repeats were performed.

### 2.6. Three-Dimensional In Vitro Experiments

#### 2.6.1. Preparation of Tumor Spheroids—Hanging Drop Method

For fluorescence microscopy experiments spheroids were generated using the hanging drop method. Briefly, small volumes (25 µL) of cell suspension (2 × 10^5^ cells/mL) were placed onto the bottom of a µ-Dish (50 mm, low, uncoated) from ibidi GmbH (Martinsried, Germany) and the dish was inverted afterward forming “hanging drops”; 500 µL of Dulbecco’s Phosphate Buffered Saline, DPBS (no calcium, no magnesium; Gibco^®^, Thermo Fisher Scientific, Waltham, MA, USA) were placed into the lid of the µ-Dish to create a hydrated atmosphere.

#### 2.6.2. Visualization of Cell Death 3D—Fluorescence Microscopy—PI Uptake

Cell death induced by antitumor peptides on tumor cell spheroids was also analyzed by detection of PI-uptake by microscopy. After the generation of hanging drops for 72 h as described above, the peptide was carefully added at different concentrations (0–100 µM) to the hanging drops. Spheroids were then incubated in presence of peptide at 37 °C and 5% CO_2_ for 24 h. After the addition of 2.5 µL PI (10 µg/mL), spheroids were incubated for another 10 min before microscopic images were taken. Experiments were performed with a Leica DMI6000 B with IMC using a Leica DFC360 FX camera and AF 6000 software (Leica Microsystems, Vienna, Austria). Excitation and emission wavelengths were 536 nm and 617 nm, respectively.

#### 2.6.3. Generation of Tumor Spheroids—Ultra-Low Attachment Plates

The MUG-Mel2 cells were seeded at 10^4^ cells per 100 µL in the respective medium into black Corning^®^ 96-well Spheroid Microplates with a round clear bottom (Ultra-Low Attachment surface) from Sigma Aldrich (Deisenhofen, Germany) and grown for 5 days at 37 °C and 5% CO_2_. About 60% of the medium was partially exchanged every second day.

#### 2.6.4. Three-Dimensional Viability Assay

The CellTiter-Glo^®^ 3D-Cell Viability Assay from Promega (Promega, Madison, WI, USA) was used to determine the cell viability in 3D MUG-Mel2 spheroids prepared with Ultra-Low Attachment plates. The 3D assay reagent measures ATP as an indicator of viability and generates a detectable luminescent readout. For the assay, the spheroids were cultivated and seeded as described above and treated with peptide for 48 h. In addition, a positive control was treated with 2 µL of Triton-X-100 (10%) representing 0% of cell viability (V_0_). The luminescence signal was measured using the GloMax^®^ Discover Microplate Reader (Promega, Madison, WI, USA) at room temperature for peptide concentrations from 0 to 100 µM. The plate was shaken vigorously for 5 min and afterward, incubated for 25 min. The viability was calculated from the percentage of viable cells without peptide (V_100_) and in the presence of peptide (V_X_) (Equation (2)).
(2)$$\%\mathrm{viability}=100\times \frac{\left(V_{X}-V_{0}\right)}{\left(V_{100}-V_{0}\right)}$$

### 2.7. Human 3D In Vitro MUG-Mel2 Melanoma Organotypic Models (mOS-REp^MUG-Mel2^)

#### 2.7.1. General Cell Culture

According to a previously published protocol [22], primary human epidermal keratinocytes were isolated from foreskin biopsies from juvenile donors between 3 and 6 years.

Keratinocytes were cultured in E1 medium (EpiLife^®^ medium supplemented with 1% human keratinocytes growth supplements (HKGS) and 1% penicillin/streptomycin (all Thermo Fisher Scientific, Waltham, MA, USA)). MUG-Mel2 was cultured in RPMI medium (Thermo Fisher Scientific, Waltham, MA, USA) containing 10% fetal calf serum (FCS; Bio & Sell, Nürnberg, Germany) and 1% penicillin/streptomycin (Thermo Fisher Scientific, Waltham, MA, USA).

#### 2.7.2. Generation of Melanoma Open Source Reconstructed Epidermis Models (mOS-REp^MUG-Mel2^)

Melanoma organotypic models were generated according to a previously published protocol [19]. Briefly, 5 × 105 keratinocytes at passage 3 were mixed with a pre-defined ratio of melanoma cells in E2 medium (E1 medium supplemented with 1.44 mM CaCl2) and seeded on a polycarbonate membrane (pore size 0.4 µm) with 12-well inserts (Greiner Bio-One, Frickenhausen, Germany). After 24 h, the models were set to the air-liquid-interphase and the medium was changed to E3 medium (E2 medium containing 10 ng/mL Keratinocyte Growth Factor (Thermo Fisher Scientific, Waltham, MA, USA) and 73 μg/mL ascorbin-2-phosphat). For a culture period of 21 days, the medium was changed three times per week.

#### 2.7.3. Treatment of Melanoma Models with Peptide RDP22

Fifty grams of peptide RDP22 were applied topically on day 1 and day 2, respectively.

#### 2.7.4. Non-Destructive Fully Automated Detection of Tumor Progression and Regression

Tumor progression and regression were non-destructively detected using a fully automated, in-house developed MediTOM device. Images were taken with the transmitted light microscope and analyzed with ImageJ (Rasband, W.S., ImageJ, U. S. National Institutes of Health, Bethesda, MD, USA).

#### 2.7.5. Determination of Cellular Integrity

Cellular integrity was analyzed photometrically in cell culture supernatants using the Cedex Bio Analyzer (Roche, Königswinter, Germany). Values were determined by measurement of the concentration of the intracellular protein Lactate dehydrogenase (LDH) indicative of the rupture of the cell membrane.

#### 2.7.6. Assessment of Metabolic Activity

To analyze the metabolic activity of the models, a 3-(4,5-dimethyldiazol-2-yl)-2,5-diphenyltetrazolium bromide (MTT) assay was performed on day 21. Tissues were incubated in 1 mg/mL MTT (Merck, Darmstadt, Germany) at 37 °C for 3 h. After extraction of formazan salt from the models using 2 mL 2-propanol, a quantitative analysis via absorbance measurement at 570 nm using a spectrophotometer (Infinite 200 M; Tecan, Männedorf, Switzerland) was conducted. Values were normalized to the untreated replicates of each group.

#### 2.7.7. Histological Assessment

For histological assessment, 4 µm histological cross-sections were prepared from tissue samples which were fixed in 4% paraformaldehyde and embedded in paraffin. Hydrated cross-sections were stained with hematoxylin and eosin (HE) for a morphological overview of the models. For immunofluorescence analysis, hydrated cross-sections were blocked using 5% BSA in phosphate-buffered saline (Merck, Darmstadt, Germany) containing 0.2% TritonTM X-100 (Merck, Darmstadt, Germany) for 20 min at room temperature. Subsequently, slides were incubated with primary antibodies against Ki67 (recombinant monoclonal rabbit anti-Ki67 (Cat# ab16667, 1:100, Abcam, Cambridge, United Kingdom)) and HMB45 (monoclonal mouse anti-melanosome/HMB-45 (Cat# M0634, 1:50, Agilent Dako, Santa Clara, CA, USA)) overnight at 4 °C. Slides were washed and incubated with fluorophore-conjugated secondary antibody (polyclonal donkey anti-mouse Alexa Fluor 647 (Cat# A31571) or polyclonal donkey anti-rabbit Alexa Fluor 555 (Cat# A31572), respectively, both 1:400, Thermo Fisher Scientific, Waltham, MA, USA) for 1 h at room temperature. After repeated washing, slides were covered with Fluoromount-DAPI (Thermo Fisher Scientific, Waltham, MA, USA).

### 2.8. In Vivo Experiments

CR ATH HO mice (Crl:NU(NCr)-Foxn1nu, Charles River Laboratories, Kent, UK) were maintained in-house (4–5 weeks of age, weight between 15 and 20 g, 5 in each group). In accordance with a protocol approved by the committee for institutional animal care and use at the Austrian Federal Ministry of Science and Research (BMWFW) (vote 66.010/0046-WF/V/3b/2016), animal work was carefully carried out. All mice were maintained under specific pathogen free (SPF) conditions in individually ventilated cages with ad libitum access to food and water. For cell injection and tumor volume measurement, mice were anesthetized with constant administration of 2% isoflurane in a constant airflow of 2.5 L/min (2.5 × 10^6^ cells in 100 µL PBS). Inoculation of the tumor was monitored daily by using a caliper, starting on day eight post-injection. After sacrifice, a histopathological examination was performed. The mice were dissected and the tumors were extracted. All other organs were checked visually for structural changes. The tissue was then fixed in a 4% paraformaldehyde solution for 24 h and further embedded in paraffin. After embedment in paraffin, four-micron sections were stained with hematoxylin and eosin. Immunohistochemical studies were performed in three cases of control and three peptide treated cases to evaluate the proliferative activity. Therefore, Ki67 staining was performed by using the anti-human Ki67-antigen clone MIB1, with EnVision FLEX (Agilent Dako Omnis, Santa Clara, CA, USA).

### 2.9. Statistical Analysis

Values are presented as the mean ± SEM (standard error of the mean). Cell culture and microscopy studies were repeated at least three times. For microscopy studies, a set of data being representative of the respective results is shown. Where applicable, analyses of organotypic and in vivo studies were performed by applying the unpaired student’s *t*-test, where differences with *p*-values < 0.001 were considered statistically significant.

## 3. Results

### 3.1. The Melanoma Metastasis Cell Line MUG-Mel2 Carries the Tumor Marker and Peptide Target PS

The cell line MUG-Mel2 originated from a cutaneous primary, ulcerated melanoma metastasis [8] (Appendix A).

Since shown recently, that contrary to melanocytes, primary cultures (MMB, melanoma metastasis to the brain) and cell lines of melanoma (SBcl-2) and their metastases (WM9, WM164) expose a significant level of the negatively charged lipid phosphatidylserine (PS) [17], that can be a target for cationic amphipathic peptides, we now investigated the cell line MUG-Mel2 for potential exposure of the tumor biomarker (Figure 1). Indeed, MUG-Mel2 shows two times higher PS exposure on the outer leaflet of its plasma membranes than normal human dermal fibroblasts (NHDF), offering a potential target for cationic amphipathic peptides. 

### 3.2. Antitumor Peptide RDP22 Strongly Interacts with PS in Model Systems

The specific interaction of RDP22 with the cancer model system PS has already been shown in previous studies as membrane perturbation (differential scanning calorimetry), membrane permeabilization (ANTS/DPX leakage) or surface neutralization (zeta potential) [12,13,14]. Within the present study, further interaction with PS liposomes via microscopic inspection was investigated.

With the help of fluorescently labeled RDP22, ((5-6)-FAM-) RDP22, interaction with PS was confirmed (Figure 2), indicated by co-localization of PS (Annexin V Alexa 350, blue) and peptide RDP22 (((5-6)-FAM-) RDP22, green).

### 3.3. RDP22 Selectively Kills MUG-Mel2 2D In Vitro

The cytotoxic activity of the peptide RDP22 towards MUG-Mel2 and normal human dermal fibroblasts NHDF [9] was determined by measurement of PI uptake, which is an indicator for the loss of cell membrane integrity and the induction of cell death (Figure 3B). Cells were incubated in respective media for 8 h in the presence of peptide. RDP22 kills around 85% of MUG-Mel2 cells at 20 µM, whereas less than 5% of dermal fibroblast cells are harmed. Upon treatment with different peptide concentrations, the LC_50_ values gained by analysis of PI uptake (induced cell death) were 8.5 μM for melanoma MUG-Mel2 and 63.6 μM for the non-neoplastic fibroblasts NHDF [9] leading to a specificity for melanoma of 7.5-fold.

Further supporting results were obtained through the microscopic inspection of PI-uptake over time in MUG-Mel2 melanoma cells. RDP22 was therein shown to exert severe toxicity on the melanoma cell line MUG-Mel2 upon 8 h of incubation and 20 µM peptide concentration (Figure 3A).

Regarding the killing mechanism, it has been shown for melanoma Sbcl2 [13] and A375 [11,12], that active cell death in the form of apoptosis occurs induced upon incubation with RDP22 dependent on PS exposed by the melanoma cells [12]. This mode of killing was also confirmed within the present study in MUG-Mel2 (see Figure 4). In the absence of peptide RDP22, MUG-Mel2 shows moderate but significant PS exposure, as it is already elevated (see Figure 1 and 4(A1)). This is indicated by the binding of Annexin V to the outside of the plasma membrane, if intact (green staining). This significantly differs from the lack of PS exposed by non-cancer melanocytes (FOM-1) or dermal fibroblasts (NHDF) as reported recently [17]. MUG-Mel2 cells in the absence of peptide were further shown to be non-apoptotic and non-necrotic, indicated by lack of morphological changes according to the two types of cell death; these cells had constantly low, but significant PS-levels exposed, no morphological changes, a lack of apoptotic blebbing, a lack of PI-uptake and significantly reduced caspase 3/7 activity compared to peptide treated cells (Figure 4(A1,C)). This is again in correlation with reports of cancer-specific PS exposure in the absence of ongoing apoptosis [17].

Upon incubation with 5–20 µM of peptide for one to six hours, PS exposure as a sign of apoptosis increases significantly in the MUG-Mel2 cells, indicated by a strong increase in the binding of Annexin V to the outside of the cell membrane, visualized as bright green spots by microscopy (Figure 4(A2,A3)), or an increase in luminescence signal, observed by spectroscopy (Figure 4(B1–B3)). Further, signs of peptide-induced apoptosis appear in form of morphological changes such as rounding, cell shrinkage and release of membrane blebs. Apoptotic cell death then decreases at higher peptide ratios up to 20 µM, whereupon cell death by late apoptosis or secondary necrosis slightly increases, indicated by PI-uptake (Figure 4(A3), red staining, arrow; Figure 4(B3), red circles). In these cells, peptide-induced membrane damage also allows the binding of Annexin V to PS in the inner leaflet of the plasma membrane. At 10 µM peptide caspase activity was increased by about 40% to 1.43-fold compared to untreated cells, further confirming induction of apoptosis in MUG-Mel2 upon peptide treatment (Figure 4C).

As already shown for melanoma A375 [12], RDP22 also interacts with the PS of MUG-Mel2 with the subsequent induction of cell death (Figure 5). This is indicated by the co-localization of PS of the MUG-Mel-2 cell membrane (blue, Annexin V Alexa Fluor 350 conjugate) with the targeting peptide RDP22 (green, ((5-6)-FAM-) RDP22). The peptide is also found inside cells in the case of induced cell death (red, PI-uptake, Figure 5). To exclude the sole adsorption of RDP22 to all membrane surfaces, ((5-6)-FAM-) RDP22 was shown to induce no significant interaction with the plasma membrane of non-malignant, non-PS-exposing fibroblasts as NHDF, nor was the significant entrance into the cells or cell death observed in presence of the peptide (Appendix A), confirming the essentiality of PS exposing membranes for interaction and sensitivity of target cells.

### 3.4. RDP22 Also Has the Potential to Kill MUG-Mel2 in 3D Tumor Spheroids

In recent years, 3D cell culture techniques have gained more and more interest since spheroids more properly mimic the tissue-like properties of tumors in vivo than monolayers in 2D cultures. Hence, 3Dmulticellular tumor spheroids (MCTS) represent a valuable tool for the evaluation of compounds like anticancer peptides for further in vivo studies.

To visualize the penetration capability of RDP22 into the core of MUG-Mel2 spheroids and the induced cell death, PI-uptake in MCTS of MUG-Mel2 upon peptide treatment was followed by fluorescence microscopy. Therefore, MCTS of MUG-Mel2 were grown for 72 h in a hanging drop culture and consequently treated with RDP22 at concentrations of 0 µM to 100 µM for 24 h. Figure 6A displays the extent of PI-uptake (red staining) correlating with peptide-induced cell death, which increases with peptide concentration. Pictures were taken in the middle plane area of the spheroids, to visualize the potential penetration capability of the peptides. Thereby, it was observed, that the peptide was also capable to enter the core of the tumor spheroids being highly active. MCTS without peptide treatment served as a negative control, monitoring the absence of cell death in absence of treatment.

For the quantification of the cytotoxic activity of RDP22 on MCTS of MUG-Mel2 a 3D viability assay measuring ATP (adenosine triphosphate) was used. Spheroids were, therefore, formed by the incubation of cells for 96 h in Corning^®^ 96-well Spheroid Microplates with a round clear bottom (Ultra-Low Attachment surface). After 48 h of peptide incubation, the potential impact on the viability of MUG-Mel2 spheroids was analyzed (Figure 6B); 3D studies show that RDP22 is not only able to penetrate into the tumor core but also efficiently kill MUG-Mel2 cells in the 3D tumor model resulting in an LC_50_ value of 51.2 ± 4.6 µM. The increase in peptide amount needed for killing 3D tumor complexes compared to 2D monolayers of tumor cells is reasonable not only because of the naturally higher hindrance of the entrance of the drug into the tumor core, but also the increased cell number in MCTS.

### 3.5. RDP22 Preventively Suppresses Melanoma Development in Melanoma Organotypic Models

The investigation of the effects of RDP22 on a melanoma organotypic model mOS-Rep^MUG-Mel2^ (control; ctrl) was performed to see whether in vivo studies are reasonable. Therefore, RDP22 (peptide) dissolved in 0.1% acetic acid (vehicle) was applied topically on days 1 and 2. Subsequently, metabolic activity and cellular integrity were assessed and differences in tumor progression and regression during development as well as proliferation rates after treatment were analyzed (Figure 7).

To detect the melanoma progression and regression non-destructively during the whole culture period, the in-house developed MediTOM device was applied. The device uses highly polarized light to detect microtumors within the skin. Images were taken with a transmitted light microscope during 21 days of culture duration at defined time points (d8, d10, d13, d15, d17, d20 and d21) (Appendix A) and analyzed with ImageJ (Figure 7A). The analysis revealed a 4.2 to 4.7-fold higher amount of developed melanoma in control (34.5%) and vehicle-treated models (31.2%) in contrast to models treated with RDP22 (7.3%).

To investigate the influence of the therapeutic peptide on proliferation rates and stable melanoma characteristics, melanoma skin equivalents were stained with the proliferation marker Ki67 and the melanoma marker HMB45, respectively. Subsequently, Ki67-positive cells were counted quantitatively (Figure 7B). Numerous Ki67-positive cells could be observed in untreated ctrl models. While there was no significant response to the treatment in models treated with the vehicle alone, the number of Ki67-positively stained cells decreased to 40% after treatment with 50 µg peptide.

Additionally, to analyze the metabolic activity of treated mOS-REp, a viability assay was performed directly in the model (Figure 7C). Only upon peptide treatment, was a slight reduction in the viability measured, which might be related to peptide-induced cell death of the small amount of melanoma within models treated with RDP22.

Furthermore, the cellular integrity was assessed by direct measurement of LDH release in the supernatant (Figure 7D). Increased LDH release correlates with decreased cellular integrity. Thus, the rupture of the cell membrane and cell death were observed as a consequence of peptide treatment.

Moreover, for a morphological overview of the models, cross sections were stained with hematoxylin and eosin (Figure 7E). In control and vehicle-treated models, melanoma nests were detectable physiologically at the basal layer of a well stratified epidermis and stochastically distributed across the entire model. In contrast, in models treated with RDP22 significantly fewer and smaller tumor nests were visible and the morphology of the models was altered indicated by a thicker stratum corneum.

### 3.6. RDP22 Causes Strong Tumor Regression of MUG-Mel2 in Human Mouse Xenografts

Due to the high selectivity of RDP22 for MUG-Mel2 in the model and in vitro in 2D, 3D and melanoma organotypic models, the peptide was further studied in a human xenograft. FOXN1 mice were, therefore, xenotransplanted with human MUG-Mel2. Tumors were grown to sizes of approximately 20–25 mm^2^. Mice were then treated either with buffer PBS (control mice C+, *n* = 5) or with peptide RDP22 (peptide treatment P, *n* = 8) at 11 doses within 14 days by intratumoral injection of buffer or peptide, respectively. As shown in Figure 8 the pigmented cell line MUG-Mel2 was growing well before treatment. In the control mice, injected with PBS, significant ongoing tumor growth is seen (C+ 112, representative control mouse, 11 injections PBS). P 94, a representative for peptide treatment with RDP22, shows strong clearance of the tumor site upon intratumoral injection of 11 doses of 0.2 mg, each, visible by respective reduction in pigmented cells.

Quantitative analysis of tumor sizes measured externally further confirmed the representativeness of the results shown above. Intratumoral treatment with 2.2 mg of RDP22 efficiently decreased the tumor size of MUG-Mel2 to a mean of 8.9 mm^2^ by about 4-fold compared to control tumors with the sole injection of buffer PBS (C+) with a mean size of 36.3 mm^2^ (Figure 9A). The high efficacy of the antitumor peptide is also reflected in further RDP22 treated tumors, as P 98 and P 100, besides P 94 (Figure 9(B2)) in comparison to control mice C+ 114 and C+ 116, besides C+ 112 (Figure 9(B1)). At the end of the experiment on day 14, upon peptide injection besides the disappearance of tumor cells, in some cases small wounds appeared, which, though were healing within 24 h. Again, an obvious clearance of tumor cells upon peptide treatment by the clearance of pigmentation is observable.

Histological staining (Figure 10) of the tumor biopsies taken on day 15 with HE (hematoxylin and eosin), displays tumor tissue in dark violet, revealing that peptide treatment (P; middle and bottom), induces a strong decrease in the tumor area and only small areas of the biopsies consist of tumor cells. In control tumors (C+; top), complete biopsies are comprised of tumor material. Double-headed arrows mark large tumor areas in control mice whereas single-headed arrows point to significantly reduced tumors in peptide-treated mice. In addition, staining with proliferation marker Ki67, proved a strong reduction in cell proliferation in MUG-Mel2 treated with peptide in comparison to untreated control tumors (Figure 11). The external observation of loss of pigmentation in the peptide treated tumors is therefore histologically confirmed to be in correlation with tumor regression. 

The analysis of the histological staining of the respective biopsies further confirmed the strong effect of RDP22 on tumor progression. Mean external tumor sizes (in vivo) at the start of treatment (day 0) determined by measurement of the pigmented tumor areas from the outside (Figure 12A; external, ext.; in vivo) were in the control and peptide group about 20–25 mm^2^. In the control group, they increased to about 36 mm^2^ until the end of the study on day 14, whereas in the peptide treatment group the mean external size decreased by around 65% to 8.9 mm^2^. Real (internal, int; ex vivo) tumor sizes indicated by HE-staining of biopsies revealed an even stronger reduction in tumors of about 12-fold in the peptide group to 0.19 mm^2^, compared to 2.31 mm^2^ upon injection of sole buffer in the control group. Thus, the real tumor size was, on average, decreased by the peptide by more than 90% compared to the control group (Figure 12B). Statistically analyzed differences were all found to be significant, yielding p-values below 0.001 (*, ** and ***, Figure 12).

## 4. Discussion

Within this study, the effect of a so far well-characterized antitumor peptide, RDP22 (R-DIM-P-LF11-322) on the highly aggressive melanoma metastasis cell line MUG-Mel2, carrying an NRAS mutation, was investigated. RDP22 was originally derived by elongation of the short peptide moiety LF11-322, which has shown strong antibacterial activity, but was lacking antitumor effects [14]. Since it has been reported by Yang et al. [23] that a requirement for antitumor peptides is a minimum net charge near +7 and a sequence length of at least 14 amino acids, the retro sequence of the short peptide stretch was added and separated by a proline to induce a turn for enabling the formation of a structurally stabilized peptide. Indeed, whereas LF11-322 (net charge +5; 9 amino acids) exhibited considerable antibacterial activity with a minimal inhibitory concentration for *E. coli* of 8–16 µg/mL [24] but only negligible antitumor activity (LC_50 Sbcl-2_ >100 µM), R-DIM-P-LF11-322 (RDP22) (net charge +9; 19 amino acids) showed highly increased (LC_50 Sbcl-2_ 2.5 µM) and specific antitumor activity [14]. The idea of the addition of the retro sequence was to design a ‘‘twin’’ peptide with increased amphipathicity and a higher propensity to form a stable secondary structure. The elongated di-retro-peptide RDP22 exhibited highly increased antitumor activity with a cancer-specificity of more than 1000-fold for cell lines of primary melanoma (Sbcl-2, NRAS mutated) and more than 500-fold for melanoma metastases (WM164, BRAF mutated) at a peptide concentration of 20 µM as compared to normal melanocytes [14]. Additionally, high toxicity in glioblastoma and rhabdomyosarcoma cells was shown [13]. Propidium iodide-uptake of melanoma cells upon incubation with peptide RDP22 further nicely demonstrated that the peptide operates via a membrane-mediated mechanism since propidium iodide can only be taken up by cells that suffer membrane disintegration [14]. It was, moreover, demonstrated that the peptide-induced apoptosis in SBcl-2 or A375 melanoma cells occurred due to the specific interaction with the PS exposed by their membranes and the formation of a bioactive structure to enter the cell and reach the Golgi upon hours of incubation, followed by mitochondrial swelling, cytochrome C release, lipid trafficking, apoptotic blebbing and finally apoptotic cell death [12,13]. The necessity for PS exposure was further confirmed by the conversion of PS to phosphatidylethanolamine and the consequent significant decrease in peptide activity toward melanoma [12].

Within the current study, the peptide RDP22 was now studied for its toxicity on MUG-Mel2 [8]. The cell line showed extremely aggressive and fast growth behavior, even in 3D cultures and animal models with persisting phenotypes [8]. The cell line was derived from a highly aggressive cutaneous metastasis that did not respond to any treatment, neither targeted, immune or cytotoxic therapy, and had the characteristic of persisting pigmentation [8] (Appendix A); MUG-Mel2 was chosen to be a perfect model for studies on new therapies in the form of antitumor peptides, such as RDP22. The natural pigmentation of the cell line allows for morphologic observations in in vivo models without using any staining additives.

To reduce the number of animals, a thorough sequence of model studies was now lined up in front of the in vivo experiment. Starting with studies on PS exposure of MUG-Mel2, liposomal and cellular PS-peptide interaction studies, and toxicity in vitro studies with 2D and 3D cultures and a melanoma organotypic model, the in vivo studies with a MUG-Mel2 xenograft followed.

As already shown for different cancer cell types as glioblastoma, rhabdomyosarcoma [17], breast cancer [25,26], leukemia [27] and melanoma and their metastasis [17,28], MUG-Mel2 exposes significantly increased levels of PS, two-fold compared to non-malignant cells, such as normal human dermal fibroblasts (NHDF). This negative charge on cancer cells can be discriminated by cationic amphipathic peptides, such as RDP22, from neutral non-PS-exposing plasma membranes of non-malignant cells and thereby be selectively targeted [18]. Indeed, a colocalization of PS to liposomes as well as MUG-Mel2 with the peptide RDP22 could be shown within the present study. Such a colocalization of antitumor peptide and PS was also reported for peptide D-K_6_L_9_ with PS exposed by prostate cancer cells [29].

Recently, we were able to prove that the PS exposed by melanoma cells of A375 is like a door-lock for specific entrance for peptides, such as RDP22, into the cancer cell for subsequent effective induction of cell death via apoptosis upon interaction with inner targets such as the Golgi [12]. In vitro studies with MUG-Mel2 also show high specificity of about 7.5-fold for MUG-Mel2 (LC_50 MUG-Mel2_ 8.5 µM) compared to normal dermal fibroblasts NHDF (LC_50 NHDF_ 63.6 µM) [9]. Likewise, mainly apoptosis was shown to be induced by RDP22 in the MUG-Mel2 cells. As previously reported for RDP22 apoptosis-induced cell death is correlated with cancer-specific toxicity compared to non-specific peptides such as R-DIM-LF11-318, inducing necrosis, which harms malignant and non-malignant cells [12].

To more closely mimic in vivo conditions within the study, in vitro experiments were expanded to 3D cultures. Before, we were already able to show that RDP22 induces cell death in tumor spheroids of melanoma cells of A375 and even mixtures of melanoma and normal dermal fibroblasts, mimicking the tumor with tumor stroma, whereas spheroids of normal fibroblasts were not significantly harmed at the antitumor active concentrations [9]. Capability for entrance into the core and efficient killing of MUG-Mel2 spheroids was similarly observed within the present study, which is a necessary skill of a potent antitumor agent.

Further, MUG-Mel2 [8] was used to successfully generate in vitro melanoma organotypic models, which were treated topically with the peptide or the vehicle. As shown before melanoma matured within a physiologically stratified epidermis. The persisting pigmentation of MUG-Mel2 within the in vitro models (Appendix A) enabled the non-destructive detection of tumor progression and regression throughout the whole culture duration via the in-house developed MediTOM device. Transmission light microscopy revealed a massive regression in tumor growth upon treatment with RDP22 and allowed the assessment of the efficacy of the potential antitumor therapeutic. Additionally, the studied metabolic activity and cellular integrity as well as proliferation rates after treatment revealed a selectively preventive killing of MUG-Mel2 within the 3D skin models. Although the organotypic models already adequately mimic the complex animal models and human tissue, they still disadvantageously lack important components of the tumor microenvironment, e.g., immune system components. Thus, animal studies cannot be completely replaced yet but reduced significantly and help to decide independently of species-specific differences whether animal experiments are indicated or not.

Thus, a reasonable number of mice xenografts was consequently treated with peptide RDP22, due to the successful confirmation of PS exposed by MUG-Mel2 as a potent biomarker and target for cationic, amphipathic and PS-specific peptide RDP22; the proof of interaction of the peptide with PS in the lipid model systems in present and previous studies [13,14]; the peptide-PS interaction in cellular (MUG-Mel2) systems; the effective and specific killing of so far non-treatable MUG-Mel2 in vitro in more highly developed tumor spheroids alone or integrated into organotypic models.

The in vivo anti-melanoma activity of the related peptide R-DIM-P-LF11-334 (RDP22 -F2, -F18, A375 [16]) was improved by RDP22 towards MUG-Mel2, which was shown by stronger regression of tumor growth and proliferation and massive reduction in tumor size in the mouse xenografts upon intratumoral peptide treatment. This was observable due to the advantageous fact that the tumor cells of MUG-Mel2 keep their pigmentation throughout their cultivation in vitro and even in vivo in the xenograft (Appendix A), which makes the measurement of the external tumor area more reliable. To, however, more precisely calculate the real tumor area, histology of biopsies with HE was performed. As in the previous study [16] histologically studied biopsies of control (PBS) and peptide-treated tumors of MUG-Mel2 indicated a strong reduction not only of externally observable, but also of internal real tumor size. RDP22 thereby reduced the real tumor size/area by about 90%, compared to the control tumors with solely PBS injection. In comparison, R-DIM-P-LF11-334 reduced internal tumor size by about 70% within these groups in A375 xenografts [16]. Similarly, within the present study, residual MUG-Mel-2 cells, if still present, were located in the inner core of the tumor surrounded by non-malignant freshly grown cells, whereas control biopsies not only were 75% more extensive but also completely comprised of viable tumor material. Proliferation marker Ki67 was further shown to be strongly reduced in peptide-treated tumors.

The fact that other antitumor peptides, e.g., LTX-315 [30] besides killing melanoma cells, also induce a peptide-enhanced antitumor immune reaction opens the possibility for further RDP22 studies in more highly elevated melanoma organotypic/immune cell models and murine tumor studies to observe a potentially similar stimulation of the immune system by RDP22 besides the direct apoptotic effect in the tumor. For systemic application, the proteolytic stability of the peptide has to be improved eventually by the introduction of D-amino acids [11]. A combination with immune therapies or cytostatic agents is also imaginable.

In conclusion, upon stepwise investigations of efficacy in increasingly sophisticated models, the antitumor peptide RDP22 was found to offer a new approach in the treatment of MUG-Mel2, an NRAS mutated melanoma with a high demand for curative treatment.

## 5. Patents

Antitumor Peptides: PCT/EP2014/050330; US 14/760,445; EP 14700349.5.

## Figures and Tables

**Figure 1 biomedicines-10-02961-f001:**
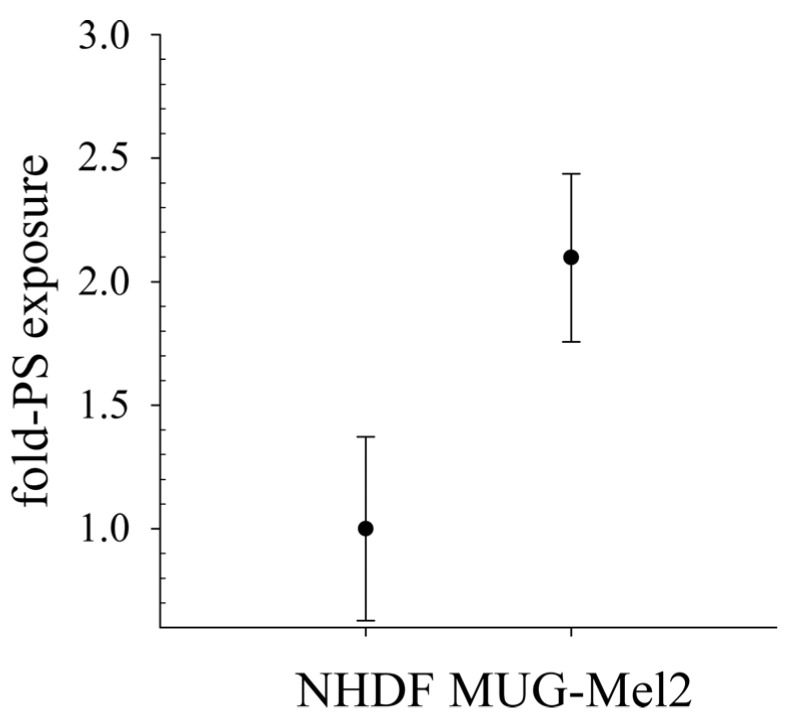
Comparison of PS exposure of normal human dermal fibroblasts (NHDF) with melanoma cell line MUG-Mel2 by measurement of fluorescence of bound Annexin V-Alexa 488. MUG-Mel2 shows significant increase in PS-levels exposed. PI uptake was measured as control to exclude binding of Annexin V to the inner leaflet of plasma membranes due to necrosis. Mean values (black dots) with standard deviations of 4–10 independent experimental data are shown. Respective PS-levels exposed differ significantly (student’s *t*-test, *p* < 0.001).

**Figure 2 biomedicines-10-02961-f002:**
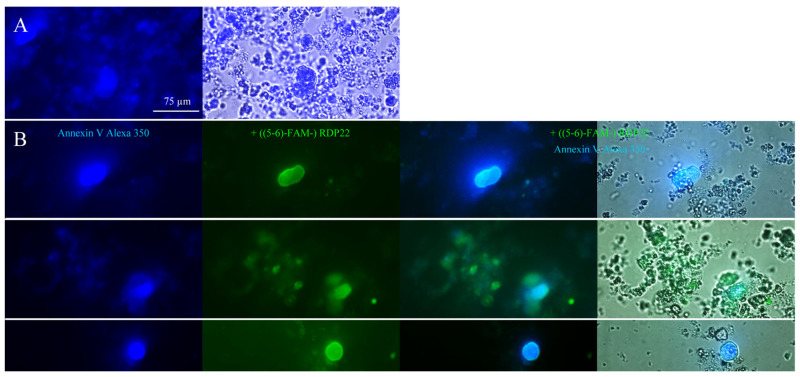
PS-peptide interaction is shown with liposomes (1–2 mM) composed of DPPS that were incubated with Annexin V Alexa Fluor 350 conjugate (blue) alone (**A**) to visualize PS (fluorescence, overlay fluorescence with bright field) or additionally with (**B**) 10 µM fluorescently labeled ((5-6)-FAM-) RDP22 (green) fluorescence, overlays of fluorescence or with bright field). Microscopy pictures shown represent the outcome of three independent experiments.

**Figure 3 biomedicines-10-02961-f003:**
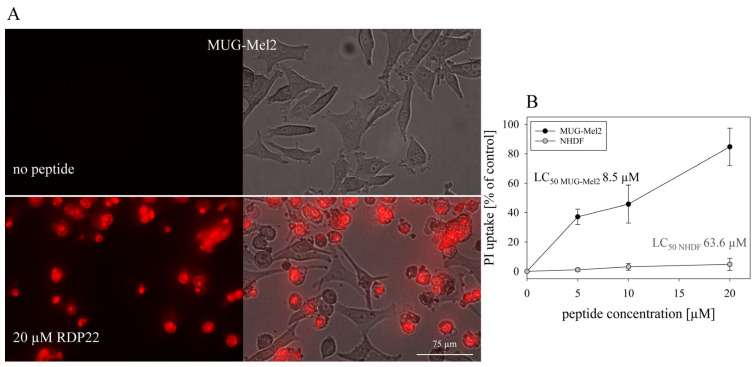
PI uptake: (**A**): Cytotoxic activity of RDP22 on MUG-Mel2 upon 8 h of incubation indicated by microscopic inspection of peptide-induced PI-uptake (red staining), which can be correlated to severe peptide-induced membrane damage and cell death. At 20 µM peptide concentration the majority of the MUG-Mel2 cells are killed. (**B**): PI-uptake and cell death of MUG-Mel2 (black circles) were also confirmed by fluorescence spectroscopy upon 8 h induced by up to 20 µM of RDP22. At the respective concentrations of RDP22, NHDF cells (gray circles) [9] are only harmed less than 5%. The data set represents the outcomes of at least three independent experiments.

**Figure 4 biomedicines-10-02961-f004:**
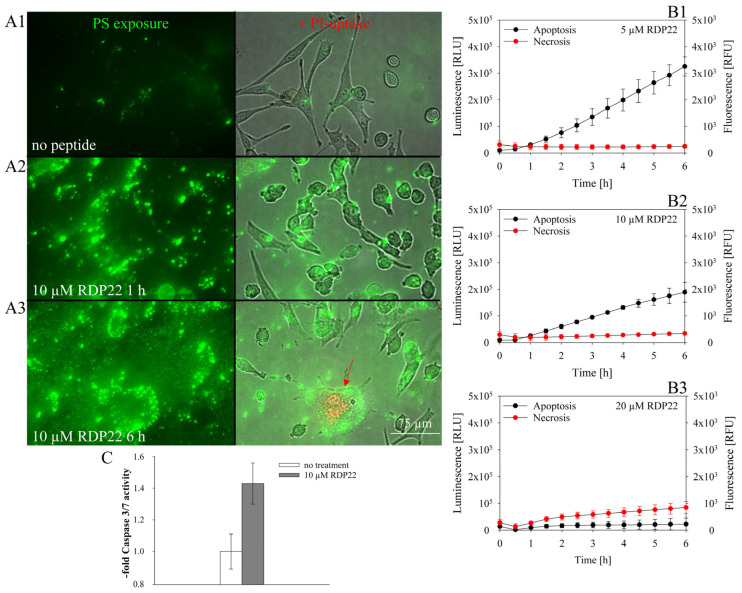
(**A**): Microscopy studies on MUG-Mel2 indicating moderate PS exposure already in the absence (**A1**) of peptide (green dye Annexin V Alexa 488). Upon 10 µM peptide incubation for 1 h (**A2**) to 6 (**A3**) hours PS exposure strongly increases indicating induction of apoptosis and upon 6 h also induction of cell death by necrosis (arrow, PI-uptake; red). Fluorescence channels for Annexin V Alexa 488 (green, PS exposure) and overlay with bright field and PI (red, cell death) are shown. Microscopy pictures represent the outcome of at least three independent experiments. (**B**): Apoptosis/necrosis induced upon incubation with peptide RDP22 at concentrations of 5 µM (**B1**), 10 µM (**B2**) and 20 µM (**B3**) was investigated. Apoptosis is shown by luminescence with black circles, necrosis is indicated by fluorescence with red circles over 6 h. From 5 to 10 µM, RDP22 induces cell death by apoptosis. By 20 µM peptide minor late apoptosis or necrosis starts to be induced. Mean values with standard deviations are derived from five independent data sets. (**C**): Fold caspase-3/7 activity of melanoma cell line MUG-Mel2 after 4 h of incubation with 0 and 10 µM of peptide RDP22, indicating killing by apoptosis. Mean values represent data from at least three independent experiments.

**Figure 5 biomedicines-10-02961-f005:**
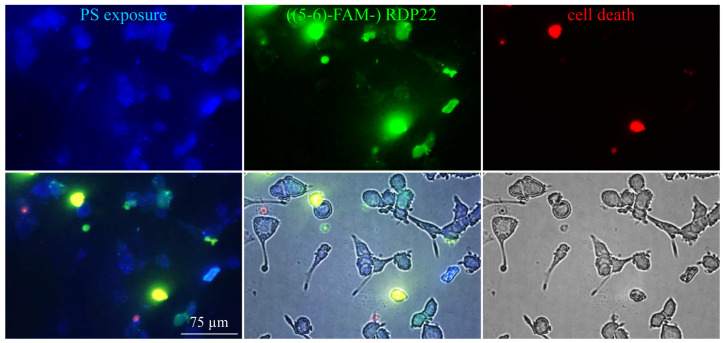
For investigation of potential PS-peptide interaction, incubation with 10 μM fluorescently labeled ((5-6)-FAM-) RDP22 (green) for 6 h was performed prior to PS-staining of MUG-Mel2 with Annexin V Alexa Fluor 350 conjugate (blue) and PI (red). In the second-row overlaps of fluorescence channels, respectively including bright field or solely bright field pictures are shown. The pictures represent outcome of 3 independent experiments. RDP22 shows interaction with PS of MUG-Mel2, as well as the whole cell area inside of dead cells.

**Figure 6 biomedicines-10-02961-f006:**
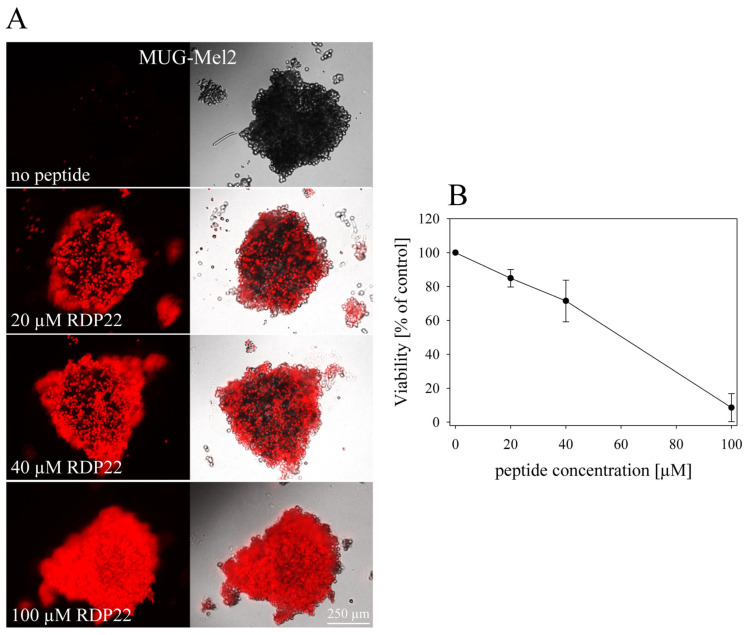
(**A**): Cytotoxic activity of antitumor peptide RDP22 on MUG-Mel2 spheroids. Spheroids were formed by incubation of cells for 72 h in a hanging drop culture. Cytotoxicity of 0–100 µM of respective peptides on different types of spheroids was determined upon 24 h of peptide incubation via PI-uptake (red). Pictures are representative of a series of several experiments and were taken in the middle plane area of the spheroids, to visualize potential penetration capability of the peptides. (**B**): 3D cell viability of MUG-Mel2 spheroids in dependence of RDP22 upon 48 h of peptide incubation from 0 to 100 µM. Spheroids were formed by incubation of cells for 5 days in Corning^®^ 96-well Spheroid Microplates with a round clear bottom (Ultra-Low Attachment surface). Untreated MUG-Mel2 spheroids were used as negative control, representing 100% of viable cells. The 3D assay reagent measures ATP (adenosine triphosphate) as an indicator of viability. Data points represent mean values with respective SD of at least three independent experiments.

**Figure 7 biomedicines-10-02961-f007:**
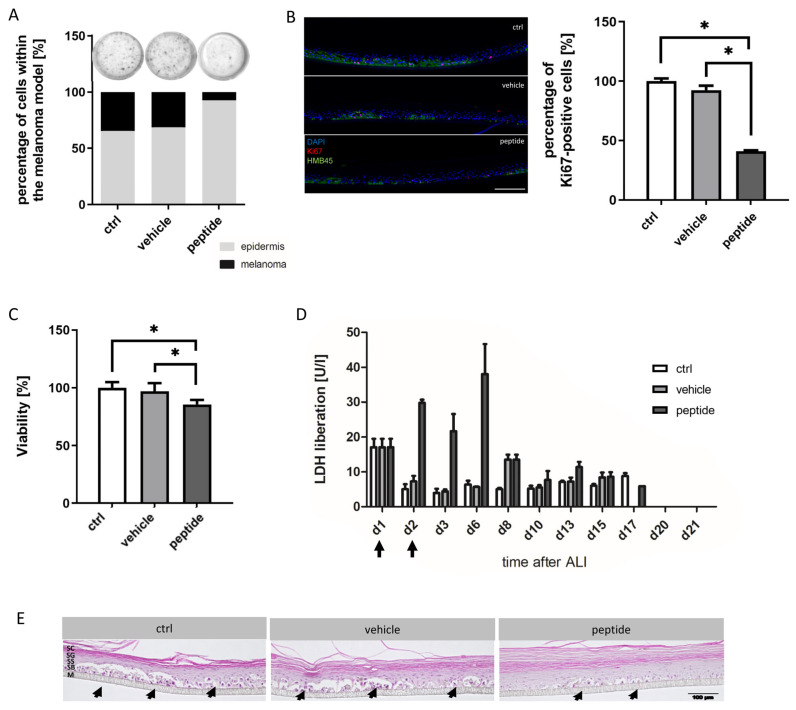
Quantification of the effects of RDP22 on mOSREp^MUG-Mel2^. (**A**): Non-destructive detection of melanoma progression and regression. Images that were taken during 21 days of culture duration (Appendix A) using the MediTOM device (upper row) were analyzed with ImageJ (lower row). In the control models and models treated with the vehicle only, a comparable amount of melanoma development could be detected. In models treated with RDP22, less melanoma growth and viability were detectable. (**B**): Influence on proliferation rates. Melanoma skin equivalents were stained with the melanoma marker HMB45 and the proliferation marker Ki67; subsequently, Ki67-positive cells were counted quantitatively. In models treated with the peptide, the number of positively stained cells decreased after treatment, whereas the Ki67 expression was not altered by vehicle treatment in comparison to the untreated control. (**C**): Analysis of metabolic activity. Melanoma skin models treated with the vehicle showed no significant difference in viability. In comparison in models treated with the therapeutic peptide, a weak decrease in viability was measurable after peptide treatment. (**D**): Determination of cellular integrity. LDH release was measured in the supernatant at defined time points during the whole culture duration. Arrows indicate the days where 0.1% acetic acid or RPP22 was topically applied to the melanoma models. LDH concentrations increased treatment-dependently in models treated with the peptide. (**E**): Hematoxylin and eosin staining of cross-sections of melanoma skin models. While smaller and fewer melanoma nests (arrows) are detectable within a morphologically altered model upon peptide treatment, in the control and vehicle-treated models a well-stratified epidermis with physiologically grown tumor nests was observed. SC = stratum corneum, SG = stratum granulosum, SS = stratum spinosum, SB = stratum basale, M = membrane. Melanoma nests are indicated by arrowheads. Scale bar 100 µm. Data are representative of two independent experiments with triplicates for each condition and represent mean values with respective SD. For (**B**) and (**C**), the student’s *t*-test was applied, showing significant differences between ctrl, respectively, vehicle and peptide treatment, *p* < 0.001 (*).

**Figure 8 biomedicines-10-02961-f008:**
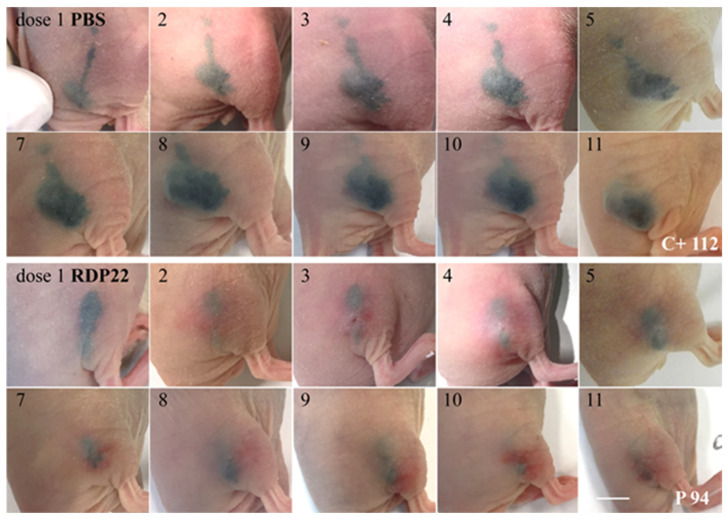
The melanogenic cell line MUG-Mel2 [22] was xenotransplanted subcutaneously in the right flank of FOXN1 mice C+ 112 (control PBS) and P 94 (peptide treatment). Before treatment, the tumors were grown to an approximate area of 20–25 mm^2^. Top rows: Representative control mouse C+ 112 was injected intratumorally with 100 µL buffer PBS 11 times within 14 days. Numbers indicate doses injected. Tumor growth of MUG-Mel2 is clearly visible due to intense pigmentation of the melanoma cells. Bottom rows: Representative peptide-treated mouse P 94 (*n* = 8) was injected intratumorally with 11 peptide doses of RDP22 of 0.2 mg, each. A total peptide amount of 2.2 mg RDP22 was used. A clear killing of MUG-Mel2 cells is observable upon peptide treatment by the strong reduction in the pigmented tumor cells. Respective doses are elevated. The bar represents 0.5 cm.

**Figure 9 biomedicines-10-02961-f009:**
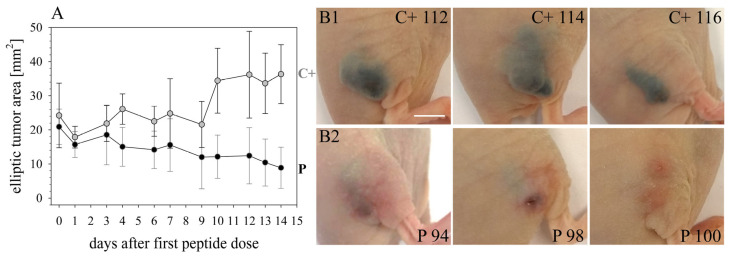
As described in Figure 8, FOXN1 mice were xenotransplanted with the melanogenic cell line MUG-Mel2 subcutaneously in the right flank. Tumors were grown to an approximate size of 20–25 mm^2^. Then injection of either 100 µL buffer PBS in control mice (C+, *n* = 5) or 0,2 mg peptide (dissolved in 100 µL PBS) in peptide-treated mice (P, *n* = 8) started at 11 doses at days 0, 1, 3, 4, 6, 7, 9, 10, 12, 13 and 14. (**A**): Peptide treatment (P) (black circles) with RDP22 strongly and significantly decreases the average tumor size by about 4-fold compared to control tumors (C+) (grey circles). The data points given reflect 11 dosages applied. (**B1**): Representative control mice C+ 112, C+ 114 and C+ 116 injected with only buffer PBS show large tumor sizes at day 14. MUG-Mel2 is clearly visible due to intense pigmentation of the melanoma cells. (**B2**): Representative peptide-treated mice P 94, P 98 and P 100 injected intratumorally with 11 peptide doses of RDP22 (2.2 mg in total) showed strong reduction up to complete clearance of MUG-Mel2 at the end of treatment at day 14, observable by the disappearance of the pigmented tumor cells. Bar represents 0.5 cm.

**Figure 10 biomedicines-10-02961-f010:**
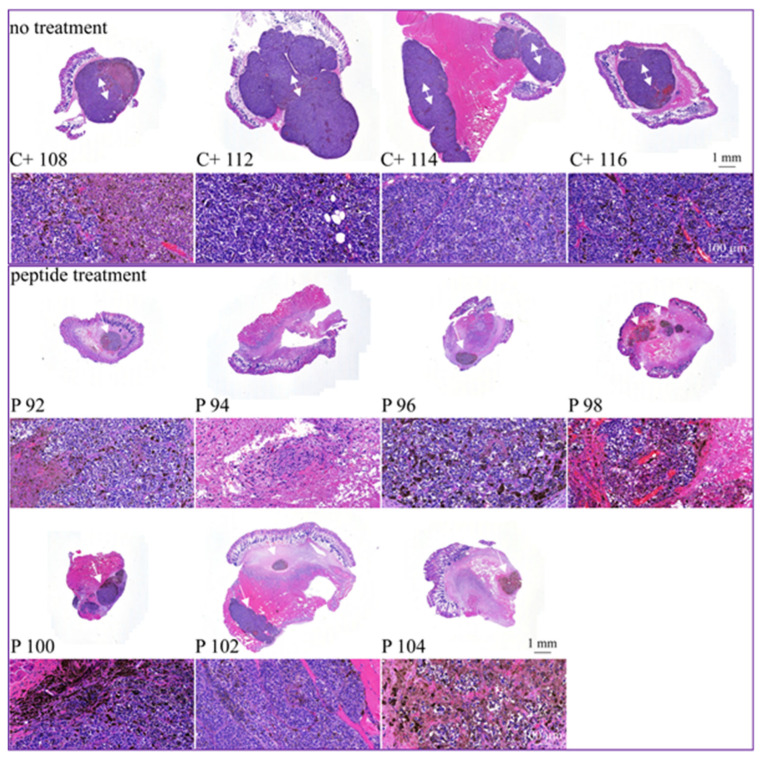
Histological staining of tumors of MUG-Mel2 xenografts in either tumors of control (C+, control PBS; top) C+ 108, C+ 112, C+ 114 and C+ 116 or peptide-treated (P, peptide treated; middle and bottom) P 92, P 94, P 96, P 98, P 100, P 102 and P 104. On day 15 of treatment mice were sacrificed and tumors were isolated and embedded in paraffin. Respective sections were stained with hematoxylin and eosin and tumor areas (dark violet, white double-headed arrows surrounded by tumor, white single-headed arrows pointing at tumor) were analyzed. Below whole tumor sections, 20-fold zooms are illustrated. Peptide treatment induces strong tumor regression with only small residual tumor areas, whereas in control tumors, tumor cells cover whole biopsy tissue and tumors are significantly larger.

**Figure 11 biomedicines-10-02961-f011:**
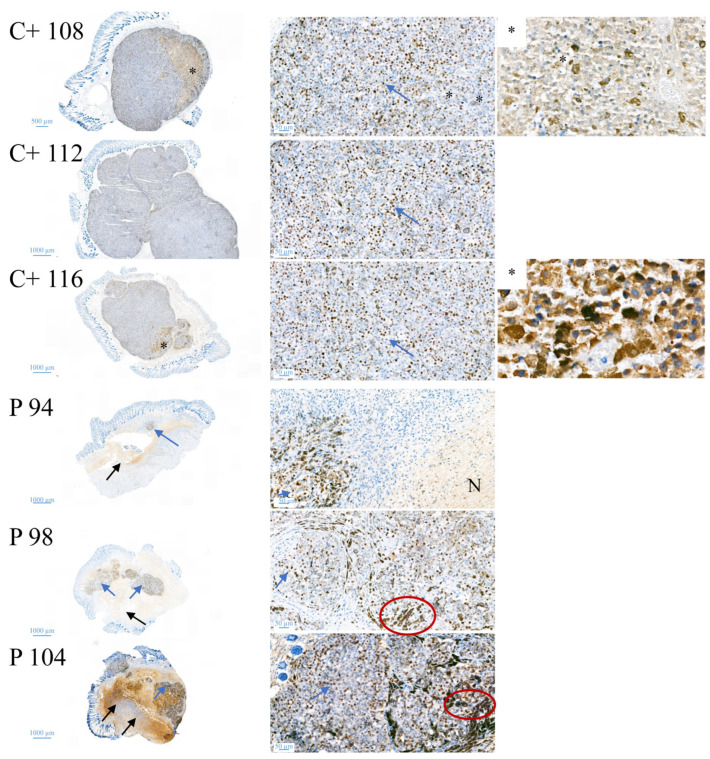
Changes in proliferation upon peptide treatment. For control tumors C+ 108, C+ 112, C+ 116 and peptide-treated tumors P 94, P 98 and P 104 proliferation was studied by Ki67 staining indicating besides strong reduction in tumor size also a strong reduction in proliferation of MUG-Mel2 upon treatment with RDP22. C+ 108, 112, 116: Untreaded vehicle control shows uniform vital tumor tissue, hardly any tumor necrosis (overview left image). C+ 108: High power view (right image): Specific nuclear Ki67 expression in approximately 35% of tumor cells (blue arrow); tumor area showing heavily pigmented tumor cells (*). C+ 112: High power view (right image): Specific nuclear Ki67 expression in approximately 35% of tumor cells, single cells with clear nuclear dark brown reactivity against Ki67 antibody (blue arrow); C+ 116: High power view (right image): Specific nuclear Ki67 expression in approximately 35% of tumor cells (blue arrow); tumor area showing heavily pigmented tumor cells (*). P 94, 98, 104: Areas of tumor necrosis with unspecific cytoplasmic staining (black arrows); areas with vital tumor cells partly heavily pigmented (blue arrows). P 94: High power view (right image): unspecific IHC staining in an area of tumor necrosis (N) small focus of vital tumor tissue with specific nuclear Ki67 expression in approximately 15% of tumor cells (blue arrow); P 98: High power view (right image): cytoplasmic melanin pigment in tumor cells (red circle); Specific nuclear Ki67 expression in approximately 20% of tumor cells (blue arrow); P 104: High power view (right image): cytoplasmic melanin pigment in tumor cells (red circle); Specific nuclear Ki67 expression in approximately 30% of tumor cells (blue arrow).

**Figure 12 biomedicines-10-02961-f012:**
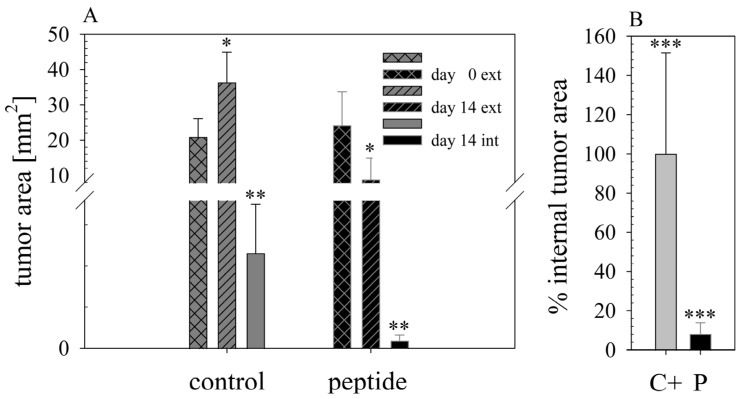
(**A**): Tumor areas, measured externally (in vivo), of control (*n* = 5) and peptide group (*n* = 8) are compared on day 0 upon start of treatment and day 14 at the end of treatment with 11 doses of buffer or peptide RDP22 (2.2 mg in total), respectively. Tumor regression is significant in the peptide group. Upon day 14 histological staining allowed determination of internal (real, ex vivo) tumor size of biopsies. Again, peptide treatment induced strong and nearly complete regression of tumors. On the y-scale (tumor area) a break was inserted from 3.6 to 8 mm^2^ for clearer comparison of all sizes. Further the lower area was enlarged for better visualization of the effects. (**B**): Percentage of internal tumor area is reduced by >90% in the peptide group. *, ** and ***: Respective tumor sizes differ significantly (student’s *t*-test, *p* < 0.001).

## Data Availability

The data presented in this study are available on request from the corresponding authors. They are not publicly available due to non-standard formats.

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
