# Peer review of "In Model, In Vitro and In Vivo Killing Efficacy of Antitumor Peptide RDP22 on MUG-Mel2, a Patient Derived Cell Line of an Aggressive Melanoma Metastasis"

_biomedicines, 2022, doi:10.3390/biomedicines10112961_

Round 1

Reviewer 1 Report

The proposed paper investigates the therapeutic effect of the RDP22 peptide in 2D, 3D and in vivo models of the aggressive NRAS-mutated MUG-Mel2 metastatic melanoma cell line. 

The topic is of clear interest in the melanoma fields as NRAS-mutated tumors need therapeutic improvements. It's a clear originality of the projet to explore peptides with anti tumor properties . 

However,  

1/ the data are based on one cell line only. The same experiments should be applied to other NRAS-mutated cell lines to confirm the significance of the results.  Could other melanoma subtypes be of interest for this new therapeutic approach by peptides (triple wt?). The group could compare more than one melanoma cell lines because of previous works (ref 13). They could easily add some control conditions. 

2/ A more relevant control for RDP22 specific cytotoxic effect , could be an "irrelevant" peptide, known to have no anti-tumor properties (LF11-322?). 

3/ it's confusing to use Annexin V Alexa apoptosis section kit for exploring  PS exposure only.  Is it a link between increased PS exposure in cancer cells and a more apoptotic phenotype? 

4/ the effect of the peptide on tumor cell death by apoptosis or necrosis is not clearly explained. Is the mechanism of death an active process?  Other molecular consequences of apoptosis should be tested (caspases cleavages, DNA fragmentation) as PS exposure is preexisting to treatment . 

5/ are the co-localizations with labeled molecules on microscopy sufficient proofs for RDP22 specific fixation on PS? How could be exclude passive adsorption of RDP22 on cell surfaces ?  Could "irrelevant" labeled-peptides be used as a control of RDP22 specificity? 

6/ it is not obvious what the 3D model add to the data.

- PI uptake is only measured by microscopy, what is the precision of the quantification of the uptake?

- 3D viability assay is difficult to compare to 2D and the mechanism of decreased viability is not documented (apoptosis linked to RPD22?). Probably, the density of cells in a spheroid can influence the viability of the spheroid. how is this managed in the test. 

7/ the melanoma organotypic model is an elegant model. But, what is really explored in figure 7B termed "metabolic activity"? 

As the differences in figure 7 B C and D statistically significant? No test is mentioned . It's not clear for figure 7   if the graph for Ki67 positive cells (no letter) and for viability (C) are linked? 

8/ As mentioned by the authors, the microenvironnement is missed in the models. Can we imagine to propose co-cultures of melanoma cell line with immune cells, to address this point with possible implication of the immune cells in the response to treatment expected in vivo?  

9/ in xenografts mice, even if the injected tumor cell lines are black, is the external size evaluation of the tumor (and his shrinking) precisely explored? Why are mentioned  biopsies analysis whereas whole tumor sections are presented in figure 10? The tumor sections treated with RDP22 or vehicle should be studied deeper with exploration of proliferation, cell death... .

The data are of clear interest but additional informations or controls are needed. The present paper needs to be improved before publication. 

Author Response

The authors thank all the reviewers for their work and detailed revision of the paper. All open questions are now answered below point by point. Besides, all changes are highlighted with track changes in the manuscript and marked in Italian below.

Due to a change in the order of citations, former Ref 22 is now Ref 8, all references in between are now +1. Reviewer 2 kindly reminded us that the former Ref 22 about MUG-Mel2 (Rinner, Sci Rep. 2017) should be listed much earlier at first mentioning.

Also, several Figures were now entitled 2A, 2B, etc. for sake of clarity. We also thank reviewer 2 for the suggestion.

Further additional experiments were performed, suggested by reviewer 1 and 2. We hope the study and manuscript is now improved and we thank the reviewers for the constructive critique.

Response to reviewers:

Reviewer 1:

Reviewer 1: Comments to the author:

The proposed paper investigates the therapeutic effect of the RDP22 peptide in 2D, 3D and in vivo models of the aggressive NRAS-mutated MUG-Mel2 metastatic melanoma cell line.

The topic is of clear interest in the melanoma fields as NRASmutated tumors need therapeutic improvements. It's a clear originality of the projet to explore peptides with anti tumor properties.

Response to Reviewer 1:

We thank the reviewer for the critical revision of the manuscript and for the constructive feedback and also for the positive feedback on the work stated above.

Reviewer 1:

1/ the data are based on one cell line only. The same experiments should be applied to other NRAS-mutated cell lines to confirm the significance of the results. Could other melanoma subtypes be of interest for this new therapeutic approach by peptides (triple wt?). The group could compare more than one melanoma cell lines because of previous works (ref 13). They could easily add some control conditions.

Response ad 1/:

We thank for the constructive comment and followed the advice and more clearly elevated other cell lines tested so far for sensitivity for RDP22 with NRAS mutation as Sbcl-2 (Riedl Biometal 2014 [14], Riedl BBA 2015 [13], Grissenberger BBA 2020 [9]). A BRAF mutated cell line WM64 also showed sensitivity for RDP22 (Riedl Biometal 2014 [14]; Riedl BBA 2015 [13]). Further, this is now mentioned in the discussion part. (since there has been a shift in reference numbers, Riedl et al BBA 2015 former [13] is now [14], etc..

Changes line 673…:

The elongated di-retro-peptide RDP22 exhibited highly increased antitumor activity with a cancer-specificity of more than 1000-fold for cell lines of primary melanoma (Sbcl-2, NRAS mutated) and more than 500-fold of melanoma metastases (WM164, BRAF mutated) at a peptide concentration of 20 µM as compared to normal melanocytes [14]……

Many of the model and in vitro (2D and 3D) studies with other cell lines (NRAS, BRAF mutated) and RDP22 were already performed by our groups and published in the above-mentioned publications. A summary is integrated in the results and discussion part, though not with the focus on mutations. RDP22 has been shown to be active for several cancer cell lines independent on any mutation, several experiments were performed in our previous work showing all in common the dependence of activity on PS exposed by multiple cancer cell lines including melanoma, A375, Sbcl-2, WM164, WM9, MUG-Mel1, MUG-Mel3, a melanoma metastasis to the brain. Thus, the focus of the work is not the NRAS mutation, though it’s a good side effect, since demanding for therapy.

Accordingly, we see, that parts of the manuscript as the introduction might be misleading that NRAS mutation is a focus in our work, whereas actually the mutation independent exposure of PS is our real focus. We are sorry for that and integrated a part in the introduction to more clearly demonstrate this fact.

Changes in line 68 and following:

PS-exposure was not only shown for different cancer types as melanoma, glioblastoma, prostate and renal cancer compared to absence in non-neoplastic control cells but it was further demonstrated that PS exposure was not attributed to ongoing apoptosis [17]. One of these peptides, RDP22 (R-DIM-P-LF11-322, PCT/EP2014/050330; US 14/760,445; EP 14700349.5), exhibits selective activity against the cancer cell marker PS in model systems and cytotoxicity against melanoma without significant effect on non-neoplastic cells as melanocytes or fibroblasts at antitumor-active concentrations [9,13,14].

As for performance of organotypic and in vivo studies with other NRAS mutated cell lines, which of course would be of great interest for us, however would unfortunately exceed the scope of our work for the current manuscript.

Reviewer 1:

2/ A more relevant control for RDP22 specific cytotoxic effect , could be an "irrelevant" peptide, known to have no anti-tumor properties (LF11-322?).

Response ad 2/:

We thank for the comment to more clearly indicate the difference between irrelevant and relevant peptides.

RDP22 is upon our findings significantly “relevant” showing high antitumor activity on MUG-Mel2 and other cancer cell lines of melanoma and glioblastoma or rhabdomysarcoma (in line 678 newly added to manuscript) ([13, 14] with negligible effect on control cells of dermal fibroblasts NHDF [manuscript and Grissenberger et al 2020; 9] or melanocytes NHEM [Grissenberger et al 2020; 9] at therapeutic concentrations. The LC50 for MUG-Mel2 is 8.5 µM and the specificity for MUG-Mel2 compared to control cells of NHDF is 7-fold. Since other control cells as, NHEM melanocytes only make up a small percentage of the human skin, only comparison with cytotoxicity on dermal fibroblasts NHDF was described herein, since considered to be more relevant. Even though, it has to be mentioned, the specificity for MUG-Mel2 over NHEM is still 2.6-fold.

The “irrelevant” peptide LF11-322 on the contrary whether shows efficient antitumor activity (addition in line 669: IC50 melanoma Sbcl-2>100 µM) nor activity on non-malignant cells (Riedl Biometal 2014; [14]) and therefore is clearly distinguishable from RDP22 (LC50 Sbcl2 2.5 µM).

This is now more clearly demonstrated by mentioning the activity of RDP22 on the third tumor cell line of rhabdomyosarcoma, and the low toxicity on tumor cells of the irrelevant peptide LF11-322 and strong differences in LC50 in the discussion:

  1. Reviewer 1:

3/ it's confusing to use Annexin V Alexa apoptosis section kit for exploring PS exposure only. Is it a link between increased PS exposure in cancer cells and a more apoptotic phenotype?

Response ad 3/:

We thank for the comment and included confirming experiments by Caspase 3/7 measurements.

We agree, that it is confusing, but the Annexin V binding is the most useful and simple tool (besides other PS-antibodies) to show PS-exposure, not only during apoptosis but also in absence of apoptosis, as e.g., reported in Riedl BBA 2011, e.g., [17], which we therein confirmed with lack of caspase cleavage, thus lack of apoptosis for several melanoma cell lines exposing PS. Thus, PS-exposure is not (only) to be attributed to an apoptotic phenotype, but is clearly a characteristic in the tumor cell membrane due to a loss of asymmetry.  

According to the comment of the reviewer, we now also more clearly confirm the statement of PS-exposure in absence of apoptosis for MUG-Mel2 by inclusion of caspase activity measurements in absence and presence of peptide (changes in line 326: Reduced caspase 3/7 activity in MUG-Mel2 cells compared to control in presence of peptide confirmed that PS-exposure is not in correlation with ongoing apoptosis (Figure 4C). line 382 and following: MUG-Mel2 cells in absence of peptide were further shown to be non-apoptotic and non-necrotic, indicated by lack of morphological changes according to the two types of cell death, constantly low but significant PS-level exposed, lack of apoptotic blebbing, lack of PI-uptake and significantly reduced caspase 3/7 activity compared to peptide treated cells) (Figure 4A1, 4C) as proposed by the reviewer in the next comment (4). Further the following was included:

Changes in line 68 and following:

PS-exposure was not only shown for different cancer types as melanoma, glioblastoma, prostate and renal cancer compared to lack of in non-neoplastic control cells but it was further demonstrated that PS exposure was not attributed to ongoing apoptosis [17].

Changes in Line 188

2.5.4. Caspase-3/7 assay

To further assess whether cells undergo apoptosis the Caspase-Glo® 3/7 Assay (Promega, Madison, WI, USA) was used. 2 x 105 cells/ml were seeded in a white 96-multiwell plate with clear bottom and grown for 48 h at 5% CO2 and 37 °C. Peptide at 0 and 10 µM was added for incubation over 4 h. Caspase solution was added with a 1:1 ratio upon 30 min at 37°C. Luminescence was measured with a GloMax® Discover Microplate Reader (Promega, Madison, WI, USA). Mean values of caspase 3/7 activity were analyzed as multiple of non-peptide treated cells. Three repeats were performed

  1. Reviewer 1:

4/ the effect of the peptide on tumor cell death by apoptosis or necrosis is not clearly explained. Is the mechanism of death an active process?

Other molecular consequences of apoptosis should be tested (caspases cleavages, DNA fragmentation) as PS exposure is preexisting to treatment.

Response ad 4/

We are sorry for missing a clearer definition of the mechanism of action, we have made some changes listed below to clarify. As suggested by the reviewer, we now in more detail describe the mechanism of cell death induced by the peptide and how apoptosis is in our experiments distinguishable from pre-existing PS exposure of the tumor cells.

Further, we included caspase measurements (in absence and presence of peptide), see also response above (ad point 3).

Line 374-387:

……. that the active cell death in form of apoptosis…………

……………….. MUG-Mel2 cells in absence of peptide were further shown to be non-apoptotic and non-necrotic, indicated by lack of morphological changes according to the two types of cell death, constantly low but significant PS-level exposed, lack of apoptotic blebbing, lack of PI-uptake and significantly reduced caspase 3/7 activity compared to peptide treated cells (Figure 4A1, 4C). ….

…………. Further, signs of peptide induced apoptosis appear in form of morphological changes as rounding, cell shrinkage and release of membrane blebs………………..

Line 399-401:

At 10 µM peptide caspase activity was increased by about 40% to 1.43-fold compared to untreated cells, further confirming induction of apoptosis in MUG-Mel2 upon peptide treatment (Figure 4C).

  1. Reviewer 1:

5/ are the co-localizations with labeled molecules on microscopy sufficient proofs for RDP22 specific fixation on PS? How could be exclude passive adsorption of RDP22 on cell surfaces ? Could "irrelevant" labeled-peptides be used as a control of RDP22 specificity?

Response ad 5/:

Unfortunately, no “irrelevant” labeled peptide is at the time available in our lab, however, we have now included data (Figure S2) on labeled RDP22 showing no interaction with membranes of non-malignant cells (human dermal fibroblasts, no PS exposure) and no induction of cell death therein, excluding sole adsorption of RDP22 to all cell surfaces and confirming the importance of PS present on the cell surface.

Accordingly new Figure S2 was added and changes were made in line 421-426:

To exclude sole adsorption of RDP22 to all membrane surfaces, ((5-6)-FAM-) RDP22 was shown to induce no significant interaction with the plasma membrane of non-malignant, non-PS-exposing fibroblasts as NHDF, nor was significant entrance into the cells or cell death observed in presence of the peptide (Figure S2), confirming the essentiality of PS exposing membranes for interaction and sensitivity of target cells.

The necessity for peptide activity on PS exposed is further underlined in the discussion in line 686 in reference to Wodlej et al. 2019 [12] in melanoma A375:

The necessity for PS exposed was further confirmed by conversion of PS to phosphatidyl-ethanolamine and consequent significant decrease of peptide activity towards melanoma [12].

  1. Reviewer 1:

6.1/ it is not obvious what the 3D model add to the data.

-PI uptake is only measured by microscopy, what is the precision of the quantification of the uptake?

Response ad 6.1/:

We are sorry for the inefficient explanation. The 3D toxicity study should be a bridge between 2D and organotypic models and further in vivo.

Changes in line 435 and following: In recent years, 3D cell culture techniques have gained more and more interest since spheroids more properly mimic the tissue-like properties of tumors in vivo than monolayers in 2D cultures. Hence, 3D- multicellular tumor spheroids (MCTS) represent a valuable tool for the evaluation of compounds like anticancer peptides for further in vivo studies.

To visualize penetration capability of RDP22 into the core of MUG-Mel2 spheroids and therein efficiently induced cell death, PI-uptake in MCTS of MUG-Mel2 upon peptide treatment was followed by fluorescence microscopy.

Compared to the 2D monolayer a 3D spheroid is physiologically more relevant and indicates in case of PI uptake into the core the peptide capability not only to kill a surface of tumor cells but also to penetrate into the tumor and kill cells therein. 

Line 445: Figure 6A displays the amount of PI-uptake (red staining) correlating with peptide induced cell death, ………... Pictures were taken in the middle plane area of the spheroids, to visualize potential penetration capability of the peptides. Thereby, it was observed, that the peptide was also capable to enter the core of the tumor spheroids being highly active. MCTS without peptide treatment served as a negative control, monitoring the absence of cell death in absence of treatment.

It is correct, that for microscopic study, only induction of cell death visualized by PI was demonstrated. Quantification was then occurring by measurement of remaining cell viability of tumor spheroids upon presence of peptide (Figure 6B). This is a necessity, since, quantification of peptide effect on tumor spheroids is experimentally difficult to handle. For quantification of PI uptake, spheroids would have to be re-separated into single cells and even then reproducible quantification would be difficult and hardly reliable. Therefore, the assay measuring ATP as indicator of viability was used, since therefore the whole spheroid is allowed to stay intact, the spheroids just swell and can be properly lysed before the measurement re-assuring minimal artificial killing, that might occur independent of peptide effects. As mentioned above, microscopic inspection was only performed to visualize capability of the peptide to enter the spheroid core (line 446: Pictures were taken in the middle plane area of the spheroids, to visualize potential penetration capability of the peptide). Thereof only a rough estimation of induced cell death is possible. Nevertheless, the quantification of cell viability is reliable.

6.2./ 3D viability assay is difficult to compare to 2D and the mechanism of decreased viability is not documented (apoptosis linked to RPD22?). Probably, the density of cells in a spheroid can influence the viability of the spheroid. how is this managed in the test.

Response ad 6.2./: We agree. As mentioned above in the response ad 6.1., the viability assay allows reproducible quantification (small error) of viability even of 3D tumors (spheroids, MCTS). For measurement of PI, spheroids would have to be disintegrated before analysis, going in hand with low reproducibility due to artificial harm of cells.

The decreased activity in 3D compared to 2D is explained in line 471: ….The increase of peptide amount needed for killing 3D tumor complexes compared to 2D monolayers of tumor cells is reasonable not only by naturally higher hindrance of entrance of the drug into the tumor core but also by increased cell number in MCTS….

We do not think that this has to be linked to a different killing mechanism as in 2D. At least the slow killing occurs upon several hours (24 hours microscopic inspection; 48 h viability assay) indicating again apoptosis to occur upon peptide incubation, which resembles killing mechanism in 2D.

That is correct, the density in the spheroid is high and depending on size and time of growth goes in hand with a partially necrotic core. To exclude non peptide induced killing, cell death with no peptide as control is analyzed (see Figure 6A no peptide, microscopy) and in case of quantification subtracted (Figure 6B) to reveal the pure peptide effect.

  1. Reviewer 1:

7/ the melanoma organotypic model is an elegant model. But, what is really explored in figure 7B termed "metabolic activity"? As the differences in figure 7 B, C and D statistically significant? No test is mentioned. It's not clear for figure 7 if the graph for Ki67 positive cells (no letter) and for viability (C) are linked?

Response ad 7/:

The authors thank the reviewer for the thorough assessment. Thank you for pointing out that we did not explain clearly enough “metabolic activity” and the link in 7C. We had to recognize that the labelling of fig. 7B and 7C was reversed resulting in a misunderstanding of the shown results. We apologize for this inconvenience and have changed these circumstances in lines 491-505 and 522-533, respectively. Melanoma skin equivalents were stained with the melanoma marker HMB45 and the proliferation marker Ki67; subsequently, Ki67 positive cells were counted quantitatively which is shown in the linked graph. To analyze the metabolic activity of the models, a viability assay with MTT was performed. Only in metabolic active viable cells MTT is transformed to blue formazan salt which can be extracted and measured using a spectrophotometer. For fig. 7B, C and D six data sets were analyzed showing the mean+SD which is now included to the figure description in line 538. In 7B and C differences between ctrl (or vehicle) compared to peptide are significant. This is also the case in 7D at day 2, 3, 6, 13 and 17.

For proving statistical significance a students t-test was performed for results of 7B and 7B indicating significance with a p-value<0.001.

Line 543: ………..condition and represent mean values with respective SD. For B and C, the student’s t test was applied, showing significant differences between ctrl, respectively vehicle and peptide treatment, p<0.001 (*).

Line 311 Statistical analysis: Where applicable, analyses of organotypic and in vivo studies were performed applying the unpaired student’s t test, where differences with p-values < 0.001 were considered statistically significant.

  1. Reviewer 1:

8/ As mentioned by the authors, the microenvironnement is missed in the models. Can we imagine to propose co-cultures of melanoma cell line with immune cells, to address this point with possible implication of the immune cells in the response to treatment expected in vivo?

Response ad 8/: The authors thank the reviewer for this constructive remark. The integration of immune cells to our models is currently beyond what is technically possible and is part of ongoing research.

9.Reviewer 1:

9/ in xenografts mice, even if the injected tumor cell lines are black, is the external size evaluation of the tumor (and his shrinking) precisely explored? Why are mentioned biopsies analysis whereas whole tumor sections are presented in figure

Response ad 9/:

It is correct, that even with the dark melanoma phenotype only a rough determination of the size is possible from the outside (external size). That is why for a more precise determination the HE staining of the biopsies was performed (Figure 12 external and internal).

By biopsy analysis we meant, measurement of the size of the dark violet HE stained tumor areas. Here, the correct tumor area can be more precisely determined than from the outside (Fig 12 int). HE staining and evaluation of tumor area is regarded a standard and commonly accepted procedure in pathology. That is why we relied on this analysis.

Line 641-645: …Real (internal; int, ex vivo) tumor sizes indicated by HE-staining of biopsies revealed an even stronger reduction of tumors of about 12-fold in the peptide group to 0.19 mm2, compared to 2.31 mm2 upon injection of sole buffer in the control group….. 

We appreciate the comment and extended the analysis of the biopsies for proliferation (see comment 10) which now improves the understanding of the peptide effect. The Ki67 staining of biopsies is now integrated as Figure 11 to confirm that not only the tumor size but also the proliferation, which is massively reduced in presence of peptide compared to the control with vehicle.

Line 601-603:

In addition, staining with proliferation marker Ki67, proved a strong reduction of cell pro-liferation in MUG-Mel2 treated with peptide in comparison to untreated control tumors (Figure 11).

Line 614-634: Figure 11. Changes in proliferation upon peptide treatment. For control tumors C+ 108, C+ 112, C+ 116 and peptide treated tumors P 94, P 98 and P 104 proliferation was further studied by Ki67 staining indicating besides strong reduction of tumor size also a strong reduction in proliferation of MUG-Mel 2 upon treatment with RDP22.

C+ 108, 112, 116: Untreaded vehicle control: uniform vital tumor tissue, hardly any tumor necrosis (overview left image),

C+ 108: High power view (right image): Specific nuclear Ki67 expression in approximately 35% of tumor cells (arrow blue); tumor area showing heavily pigmented tumor cells (*).

C+ 112: High power view (right image): Specific nuclear Ki67 expression in approximately 35 % of tumor cells, single cells with clear nuclear dark brown reactivity against Ki67 antibody (arrow blue);

C+ 116: High power view (right image): Specific nuclear Ki67 expression in approximately 35% of tumor cells (arrow blue); tumor area showing heavily pigmented tumor cells (*).

P 94, 98, 104: Areas of tumor necrosis with unspecific cytoplasmic staining (arrow black); areas with vital tumor cells partly heavily pigmented (arrow blue).

P 94: High power view (right image): unspecific IHC staining in an area of tumor necrosis (N) small focus of vital tumor tissue with specific nuclear Ki67 expression in approximately 15% of tumor cells (arrow blue);

P 98: High power view (right image): cytoplasmic melanin pigment in tumor cells (red circle); Specific nuclear Ki67 expression in approximately 20% of tumor cells (arrow blue);

P 104: High power view (right image): cytoplasmic melanin pigment in tumor cells (red circle); Specific nuclear Ki67 expression in approximately 30% of tumor cells (arrow blue).

See also next point (10.).

  1. Reviewer 1:

10? The tumor sections treated with RDP22 or vehicle should be studied deeper with exploration of proliferation, cell death....

Response ad 10/: We appreciate the comment and therefore Ki67 staining was performed, see also above point 9.

New Figure 11; line 614….: Changes in proliferation upon peptide treatment. For control tumors C+ 108, C+ 112, C+ 116 and peptide treated tumors P 94, P 98 and P 104 proliferation was further studied by Ki67 staining indicating besides strong reduction of tumor size also a strong reduction in proliferation of MUG-Mel 2 upon treatment with RDP22………..

Reviewer 1: The data are of clear interest but additional informations or controls are needed. The present paper needs to be improved before publication.

Response: We again thank the reviewer for the critical revision of the manuscript and for the constructive feedback and we hope that the changes made according to the comments now meet the required criteria for publication.

Reviewer 2 Report

In the manuscriptIn model, in vitro and in vivo killing efficacy of antitumor peptide RDP22 on MUG-Mel2, a patient derived cell line of an aggressive melanoma metastasis” Maximiliane Wußmann and colleagues describe the effect of membrane phosphatidylserine- targeting peptide PDP22 on metastatic melanoma cell line. First, peptide binding to liposomes is shown, then its effect on cultured cells, organotypic and spheroid culture is determined, with partial insight into the mechanism of peptide-triggered cell death. PDP22 kills melanoma cells in vivo, upon intratumoural injection. The observations are interesting, and peptide may be considered as a promising anti-melanoma agent. However, there are few issues to be addressed to improve the paper quality.

MAJOR REVISION:

1.     Phosphatidylserine (PS) exposure assay was used to determine the type and extent of cell death upon RDP22 application. As RDP22 itself binds PS, it may already disturb the interaction with fluorescently labelled annexin V which makes the interpretation of results difficult. Effector caspase activity assay would be more appropriate to better define cell death in the in vitro model.  

MINOR REVISION:

1.     The figures are very hard to navigate. Top right, top left etc descriptors need to be replaced with traditional Fig 1A, 1B etc.

2.     The publication on MUG-Mel2 cell line origin needs to be cited when cell line is mentioned for the first time in the introduction.

3.     The intratumoural injection of a peptide is not optimal, especially when investigated compound is supposed to work as a drug against metastatic tumour. Please explain why RDP22 wasn't applied systemically.

4.     It’s unclear why gap is introduced into y-axis (tumour area) of first graph in Figure 11 (should be Figure 11A). The normal y-axis with values ranging from 0 to 50 would fit better.

5.     Some English editing in Results part is needed.

6.     Line 144 – it is unclear whether 105 aliquots of cells were treated in the suspension or after seeding onto cell culture plates. Please specify.

Author Response

The authors thank all the reviewers for their work and detailed revision of the paper. All open questions are now answered below point by point. Besides, all changes are highlighted with track changes in the manuscript and marked in Italian below.

Due to a change in the order of citations, former Ref 22 is now Ref 8, all references in between are now +1. Reviewer 2 kindly reminded us that the former Ref 22 about MUG-Mel2 (Rinner, Sci Rep. 2017) should be listed much earlier at first mentioning.

Also, several Figures were now entitled 2A, 2B, etc. for sake of clarity. We also thank reviewer 2 for the suggestion.

Further additional experiments were performed, suggested by reviewer 1 and 2. We hope the study and manuscript is now improved and we thank the reviewers for the constructive critique.

Response to reviewers:

Reviewer 2:

Reviewer 2: Comments to the author:

In the manuscript „In model, in vitro and in vivo killing efficacy of antitumor peptide RDP22 on MUG-Mel2, a patient derived cell.  line of an aggressive melanoma metastasis” Maximiliane Wußmann and colleagues describe the effect of membrane phosphatidylserine- targeting peptide PDP22 on metastatic melanoma cell line. First, peptide binding to liposomes is shown,

then its effect on cultured cells, organotypic and spheroid culture is determined, with partial insight into the mechanism of peptide triggered cell death. PDP22 kills melanoma cells in vivo, upon intratumoural injection. The observations are interesting, and peptide may be considered as a promising anti-melanoma agent. However, there are few issues to be addressed to improve the paper quality.

Response to Reviewer 2:

We thank reviewer 2 for the critical revision of the manuscript, for the constructive feedback and also for the positive feedback on the work stated above.

Reviewer 2:

MAJOR REVISION:

  1. Phosphatidylserine (PS) exposure assay was used to determine the type and extent of cell death upon RDP22 application. As RDP22 itself binds PS, it may already disturb the interaction with fluorescently labelled annexin V which makes the interpretation of results difficult. Effector caspase activity assay would be more appropriate to better define cell death in the in vitro model.

Response ad 1: We thank for the good advice, we now integrated caspase 3/7 activity measurements to underline the results gained with Annexin V (text.

Line 399-401:

At 10 µM peptide caspase activity was increased by about 40% to 1.43-fold compared to untreated cells, further confirming induction of apoptosis in MUG-Mel2 upon peptide treatment (Figure 4C).

Line 414-416 Figure 4C. Fold caspase-3/7 activity of melanoma cell line MUG-Mel2 after 4 h of incubation with 0 and 10 µM of peptide RDP22 indicating killing by apoptosis. Mean values represent data of at least three independent experiments.

Indeed, we have the problem that PS is the peptide target and also the sign of apoptosis, therefore we always observe a competition between peptide and Annexin V, if we add the 2 components at the same time, which e.g. occurs in the apoptosis/necrosis assay (Figure 4B). Nevertheless, upon longer incubation (up to 6 hours) this appears to be less the problem, since peptide enters the cell and increase of PS exposure and apoptosis can be observed by increase of Annexin V binding as seen in Figure 4B.

For microscopic inspection peptide incubation took place before Annexin staining, no disturbance of Annexin binding occurs in that case (Figure 4A)

MINOR REVISION:

  1. The figures are very hard to navigate. Top right, top left etc descriptors need to be replaced with traditional Fig 1A, 1B etc.

Response ad 1.: We are sorry for the inconvenience; we completely understand and changed the figures accordingly.

  1. The publication on MUG-Mel2 cell line origin needs to be cited when cell line is mentioned for the first time in the introduction.

Response ad 2.: We thank the reviewer for this important observation, we changed it accordingly, the citation is now [8] instead of [22] and is cited in the introduction.

  1. The intratumoural injection of a peptide is not optimal, especially when investigated compound is supposed to work as a drug against metastatic tumour. Please explain why RDP22 wasn't applied systemically.

Response ad 3: The objection is completely justified. Systemic application will be the next step to also reach metastasis. First, however we wanted to study the direct antitumor effect independent on problems with e.g., proteolytic stability in presence of blood. For intravenous application we presumably have to stabilize the peptide by exchange of L-amino acids with D-aa or inclusion in nanocarriers.  This is an ongoing study in our lab at the moment. Further as seen in studies on another antitumor peptide, LTX315, also intratumoral application can besides direct killing of the tumor cause secondary effects on immune cells now targeting tumor cells themselves, even metastasis. This is however only possible to study with murine tumors and not in human xenografts as with MUG-Mel2.

Changes in line 780-782: For systemic application, proteolytic stability of the peptide has to be improved eventually as by introduction of D-amino acids [11].

  1. It’s unclear why gap is introduced into y-axis (tumour area) of first graph in Figure 11 (should be Figure 11A). The normal y-axis with values ranging from 0 to 50 would fit better.

Response ad 4: The reason for the gap, was that otherwise the peptide treated internal size, since so small (0.19 mm2), would not be visible any more. We tried to remove the gap, however then a comparison of the sizes in one graph would not have been possible anymore. We then, for the sake of clarity, decided to keep the gap, and hope this is ok. If not, we could compare the internal sizes in a second graph, which however prevents the observation of the differences of external and internal sizes of tumor.

Figure was changed from 11 to 12 due to integration of Ki67 studies.

We added an explanation for the gap integration in line 654: On the y-scale (tumor area) a break was inserted from 3.6 to 8 mm2 for clearer comparison of all sizes. Further all bars and error bars are visible from the bottom to the top.

We nevertheless renamed in A and B as suggested.

5 Some English editing in Results part is needed

English editing in the results part was performed.

6.

Line 144 it is unclear wether 105 aliquotes of cells were treated in the suspension or after seeding onto cell culture plates, Please specify

Response ad 6.:  cells were treated “in suspension”, this was added in line 149.

Round 2

Reviewer 1 Report

Thank you for the clear improvement of the manuscript , now suitable for publication.